



# Radiative sensitivity quantified by a new set of radiation flux kernels based on the ERA5 reanalysis

Han Huang, Yi Huang

Department of Atmospheric and Oceanic Sciences, McGill University, Montreal, Canada

Corresponding Authors:

Han Huang, han.huang2@mcgill.ca (ORCID: 0000-0002-9143-6453),

Yi Huang, yi.huang@mcgill.ca (ORCID: 0000-0002-5065-4198)



**Abstract**
Radiative sensitivity, i.e., the response of the radiative flux to climate perturbations, is essential
to understanding climate variability. The sensitivity kernels computed by radiative transfer
models have been broadly used for assessing the climate forcing and feedbacks in global
warming. As these assessments are largely focused on the top of atmosphere (TOA) radiation
budget, less attention has been paid to the surface radiation budget or the associated surface
radiative sensitivity kernels. Based on the fifth generation European Center for Medium-Range
Weather Forecasts atmospheric reanalysis, we produce a new set of radiative kernels for both the
TOA and surface radiative fluxes, which is made available at
http://dx.doi.org/10.17632/vmg3s67568.1 (H.Huang, 2022). By comparing with other published
radiative kernels, we find that the TOA kernels are in agreement in terms of global mean
radiative sensitivity and analyzed overall feedback strength. The unexplained residual in the
radiation closure tests is found to be generally within 10%, no matter which kernel dataset is
used. The inter-kernel bias-induced uncertainty, as measured by the standard deviation of the
global mean feedback parameter value, is typically no more than 10% in the longwave and 20%
in the shortwave; this uncertainty is much smaller than the inter-climate model spread of the
feedbacks. However, there exist more significant regional biases in kernel values, due to the
dependence of radiative sensitivity on the atmospheric states, and this contributes to more
significant radiation non-closure at the regional scale, such as in the Arctic and Southern Ocean
regions. On the other hand, we find relatively larger discrepancies in the surface kernels.
Although several kernels can achieve as good radiation closure compared to the TOA kernels,
affirming the validity of kernel method for the surface radiation budget analysis, the non-closure
residual in certain kernels may amount to over 100% of the total radiation change. The
intercomparison of the surface kernels reveals important biases, such as in the radiative
sensitivity to air temperature in the lowermost atmospheric layers adjacent to the surface, which
is of critical importance to the overall surface feedback strength.





## 1. Introduction

Radiative kernels measure the sensitivity of radiative fluxes to the perturbation of feedback variables, such as temperature, water vapor, albedo and cloud (e.g., Soden & Held, 2006; Y. Huang et al., 2007; Shell et al., 2008; Previdi, 2010; Zelinka et al., 2012; Block & Mauritsen, 2013; Yue et al., 2016; Y. Huang et al., 2017; Pendergrass et al., 2018; Thorsen et al., 2018; Kramer et al., 2019; Smith et al., 2020). Compared to the partial radiative perturbation method (e.g., Wetherald & Manabe, 1988), which is precise but computationally expensive, the kernel method deploys a set of precalculated radiative kernels with simple arithmetic multiplications in feedback quantification and thus is computationally highly efficient, which has greatly facilitated the analysis of radiative feedbacks in global climate models (GCM) (e.g., Soden et al., 2008; Jonko et al., 2012; Vial et al., 2013; M. Zhang & Huang, 2014; Soden & Held, 2006; Dong et al., 2020; Zelinka et al., 2020), as well as in observations (e.g., Dessler, 2010; Kolly & Huang, 2018; B. Zhang et al., 2019; H. Huang et al., 2021). These analyses have helped dissect and understand the climate sensitivity differences among the GCMs, such as those in Coupled Model Intercomparison Projects, CMIP5 (Taylor et al., 2012) and CMIP6 (Eyring et al., 2016). For example, Zelinka et al. (2020) attributed the higher climate sensitivity in the CMIP6 models to their more positive extratropical cloud feedback. The kernel-enabled feedback analyses have also provided insights in the energetics of the climate variations such as the El Nino and Southern Oscillation (ENSO, e.g., Dessler et al., 2010; Kolly & Huang 2018; H.Huang et al. 2021), the Madden-Julian Oscillation (MJO, e.g., B.Zhang et al. 2019) and the Arctic sea ice interannual variability (e.g., Y. Huang et al., 2019), despite the approximation nature of the kernel method and the known limits of its accuracy (e.g., Colman & McAvaney, 1997; H. Huang & Huang, 2021).

Multiple sets of radiative kernels have been developed to date, using different radiation codes and based on different atmospheric state datasets ranging from GCMs to global reanalysis and satellite datasets, for both non-cloud variables (e.g., Soden and Held, 2006; Shell et al., 2008; Huang et al., 2017; Thorsen et al., 2018) and cloud properties (e.g., Zelinka et al., 2012; Yue et al., 2016). As the conventional feedback analyses are mostly concerned with the radiation energy budget change at the TOA, most existing kernels have been developed and tested to address that need, i.e., to measure the feedback contributions to the TOA radiation changes. Although the radiative sensitivity depends on the atmospheric states as well as the radiative transfer codes used to compute the kernel values (e.g., Collins et al., 2006; Y. Huang & Wang, 2019; Pincus et al., 2020), it has been noted that the global mean TOA feedback quantification is insensitive to the used kernel dataset (e.g., Soden et al., 2008; Jonko et al., 2012; Vial et al., 2013). However, as there are increasing interest in regional climate change and associated feedback (e.g., Kolly & Huang, 2016; Huang et al., 2019; Zhang et al. 2019), it becomes important to know how the kernels (dis)agree at regional scales. The generation of the global radiative kernels usually requires radiative transfer computation based on a large number of instantaneous atmospheric profiles. Due to this computational cost, many kernel datasets are generated based on the atmospheric data from an arbitrary calendar year. Given the known interannual climate differences, e.g., between El Niño to La Niña years, this calls into question whether the kernels may differ in important ways for regional feedback assessments.

On the other hand, fewer feedback studies have addressed the surface radiation budget, although its importance has been recognized for such problems as the precipitation change (Previdi, 2010; Pendergrass & Hartmann, 2014; Myhre et al., 2018) and oceanic energy transport



(e.g., Zhang & Huang, 2014; Huang et al., 2017). The surface budget analysis requires the use of
surface kernels, which are not always available from the published kernel datasets. Few of them
have been subject to inter-comparisons or rigorous validation. As explained below in this paper,
the computation and use of them require different care than the TOA kernels. Possibly due to the
lack of such recognition, there exist considerable discrepancies between the existing surface
kernels and some surface budget-centered analyses reported alarmingly large non-closure in their
radiation budget analyses (e.g., Vargas Zeppetello et al., 2019), calling into question the validity
of kernel method for surface radiation budget analysis. Hence, we are motivated to examine the
radiative sensitivity quantified by different kernels, especially for the surface budget.
In this work, we produce a new set of radiative kernels for both the TOA and surface
radiation fluxes based on the fifth generation European Center for Medium-Range Weather
Forecasts atmospheric reanalysis (ERA5, Hersbach et al., 2020), which demonstrates superior
accuracy in the quantification of various atmospheric states, and document the key
considerations in the kernel computation procedure. We are interested to intercompare the
kernels computed from this atmospheric dataset to the other ones, and to investigate the
interannual variation of the kernel values due to their atmospheric state dependency. In addition,
applying a selected sets of kernels to analyzing the feedback in the CMIP6 models, we
investigate how the quantified feedback strength differs in relation to the kernel differences.

## 2. Construction of ERA5 radiative kernels


### 2.1 Radiative transfer model and atmospheric dataset

We use the GCM version of the rapid radiative transfer model (RRTMG) (Mlawer et al.,
1997) to calculate the radiative kernels. RRTMG conducts radiative transfer calculations in 16
longwave (LW) spectral bands and 14 shortwave (SW) bands. The accuracy of this model has
been extensively validated against the line-by-line calculations (e.g., Collins et al, 2006).
Input data required by RRTMG, including surface pressure, skin temperature, air
temperature, water vapor, albedo, ozone concentration, cloud fraction, cloud liquid water content
and cloud ice content, are taken from the instantaneous (as opposed to monthly mean) data of the
ERA5 reanalysis, with a horizontal resolution of 2.5 degree by 2.5 degree and 37 vertical
pressure levels. To ensure the accuracy of radiative kernels in upper atmosphere (Smith et al.,
2020), we patch five layers of the U.S. standard profile above 1hPa in the LW calculations. Other
required input variables, such as the effective radii of cloud liquid droplet and ice crystal are
taken from the 3-hourly synoptic TOA and surface fluxes and cloud product of the Clouds and
Earth's Radiant Energy System (CERES) (Doelling et al., 2013). A random cloud overlapping
scheme is used in our all-sky calculation. Sensitivity tests have been conducted to determine the
necessary temporal sampling for a proper representation of the diurnal cycle and 6-hourly and 3-
hourly instantaneous profiles are adopted for LW and SW radiative transfer calculations,
respectively, to limit the root mean squared error of the computed diurnal mean flux biases to
less than one percent.

### 2.2 Radiative kernel computation






Radiative kernels in essence measure the change of radiative flux to unit perturbation of
atmospheric variables, i.e., $\frac{\partial R}{\partial X}$, where $R$ is either the upwelling irradiance flux at the TOA or
downwelling irradiance flux at the surface; $X$ represents the aforementioned feedback variables;
$K_X$ is the radiative kernel of variable $X$. Note that for each radiative flux, $K_X$ varies with the time,
geographic and vertical locations of the perturbed variable and is in general a 4-dimensional (4-
D) data array. Note also that all radiative fluxes and kernel values are defined as downward
positive.
Following the previous studies, we compute non-cloud radiative kernels including the LW
kernels of surface temperature, air temperature, and water vapor, and the SW kernels of surface
albedo and water vapor. To calculate the kernels, we use the partial radiative perturbation
experiments, conducting two radiative transfer simulations, one without perturbation (control
run) and the other with a perturbation of one atmospheric variable. In both experiments, the
upward, downward and net radiative fluxes at the TOA and surface are saved at each time
instance and location. Then $\Delta R_0$ can be obtained by differencing the saved radiative fluxes
between the perturbed and unperturbed experiments. Dividing $\Delta R_0$ with the perturbation of
variable $X$ ($\Delta X_0$), the instantaneous radiative kernel $K_X$ is calculated as
$$K_X = \frac{\Delta R_0}{\Delta X_0} \qquad (1)$$
Applying such perturbation computations to all the relevant variables (see Appendix for a
detailed discussion of the procedure), we obtain instantaneous radiative kernels of these
dimensionalities: the surface temperature and albedo kernels are 3-D arrays (time, latitude|73,
longitude|144), and the air temperature and water vapor kernels are 4-D arrays (time, level|37,
latitude|73, longitude|144).
To account for possible interannual variability of the radiative kernel values, we compute
the kernels using atmospheric data of five calendar years: from year 2011 to 2015. Among these
years, 2011 is a strong La Niña year, 2015 is a strong El Niño year. Monthly or annual mean
kernels are then averaged from the instantaneous computations. For example, the LW annual
mean kernels of 2011 is obtained as $K = \frac{1}{365*4}\sum_{i=1}^{365*4} K_i$ and the SW kernels, $K =$
$\frac{1}{365*8}\sum_{i=1}^{365*8} K_i$, where the index $i$ represents the time slices included in the averaging. The
analyses in this work are based on multi-year mean kernels if not otherwise stated.
**3. Characterization of ERA5 kernels**
In this section, we first present the radiative sensitivity quantified by the ERA5 kernels and
compare them with the other kernel datasets. Then, we examine the interannual variability of the
ERA5 kernel values, due to the dependency of radiative sensitivity on the background
atmospheric state.
**3.1 Distribution of radiative sensitivity**

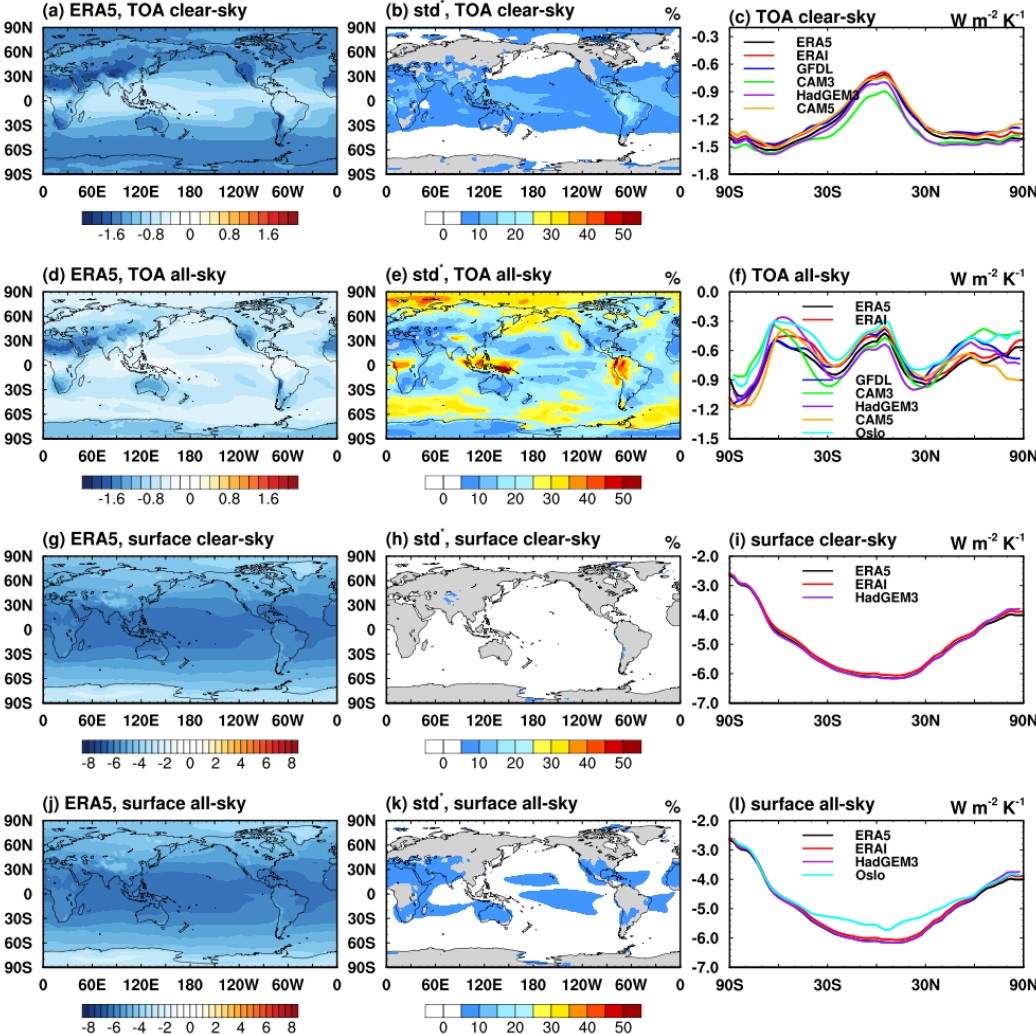

Figure 1. The sensitivity of (a-f) TOA and (g-l) surface fluxes to surface temperature,
units: W m$^{-2}$ K$^{-1}$. (a, d, g, j) The annual mean ERA5 sensitivity kernel; (b, e, h, k) the fractional
discrepancies, as measured by normalized standard deviation of the kernels listed in the right
panel in each raw; (c, f, i, l) zonal mean values of the respective kernels.
Figure 1 (left column) illustrates the ERA5 surface temperature kernels, i.e., the sensitivity
of TOA and surface fluxes to surface temperature. An increase of surface temperature leads to
more outgoing longwave radiation (OLR) both at surface and TOA, therefore the kernel is of
negative values. The TOA flux sensitivity in clear-sky (Figure 1a) is stronger than that in all-sky
(Figure 1d) due to the absence of cloud, and the value increases with latitude, due to the
decreasing concentration of water vapor from the tropics to the poles. The all-sky TOA
sensitivity is strongly influenced by clouds, showing, for example, the fingerprint of the ITCZ in
the tropical oceans. The locations with less atmospheric absorption due to less water vapor or



cloud, e.g., in the Tibetan Plateau and Sahara Desert regions, show relatively stronger sensitivity.
For the surface flux kernels, the increase of surface temperature enhances the upward emission
according to the Planck function and thus the distribution follows that of surface temperature in
both clear-sky and all-sky (Figure 1g and j).

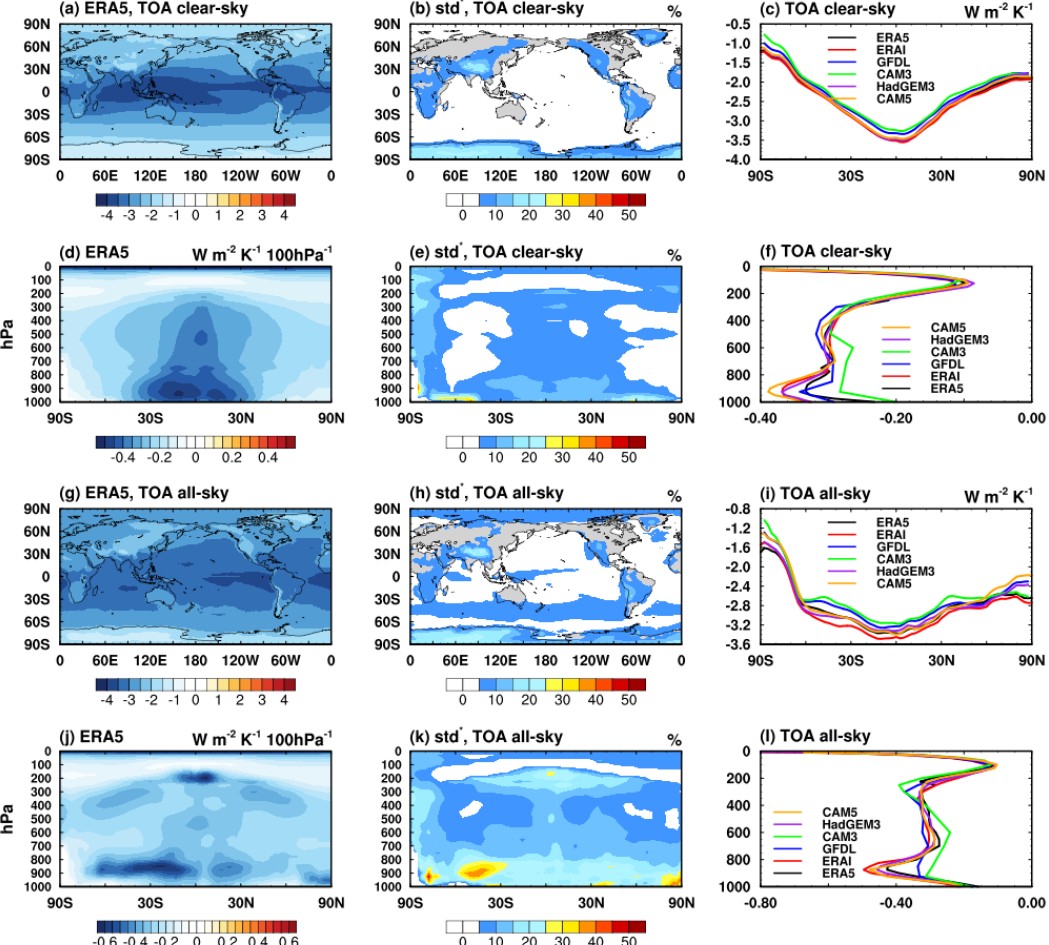

Figure 2. The sensitivity of the TOA flux to air temperature, in (a-f) clear-sky and (g-l) all-
sky. (a, d, g, j) The annual mean ERA5 kernels, among which (a) and (g) are the vertically
integrated sensitivity, units: W m$^{-2}$ K$^{-1}$, and (d) and (j) are the vertically resolved sensitivity,
units: W m$^{-2}$ K$^{-1}$ 100hPa$^{-1}$; (b, e, h, k) the fractional discrepancies of the radiative kernels; (c, i)
zonal mean vertically integrated radiative kernels, units: W m$^{-2}$ K$^{-1}$; (g, l) global mean vertically
resolved kernels, units: W m$^{-2}$ K$^{-1}$ 100hPa$^{-1}$.

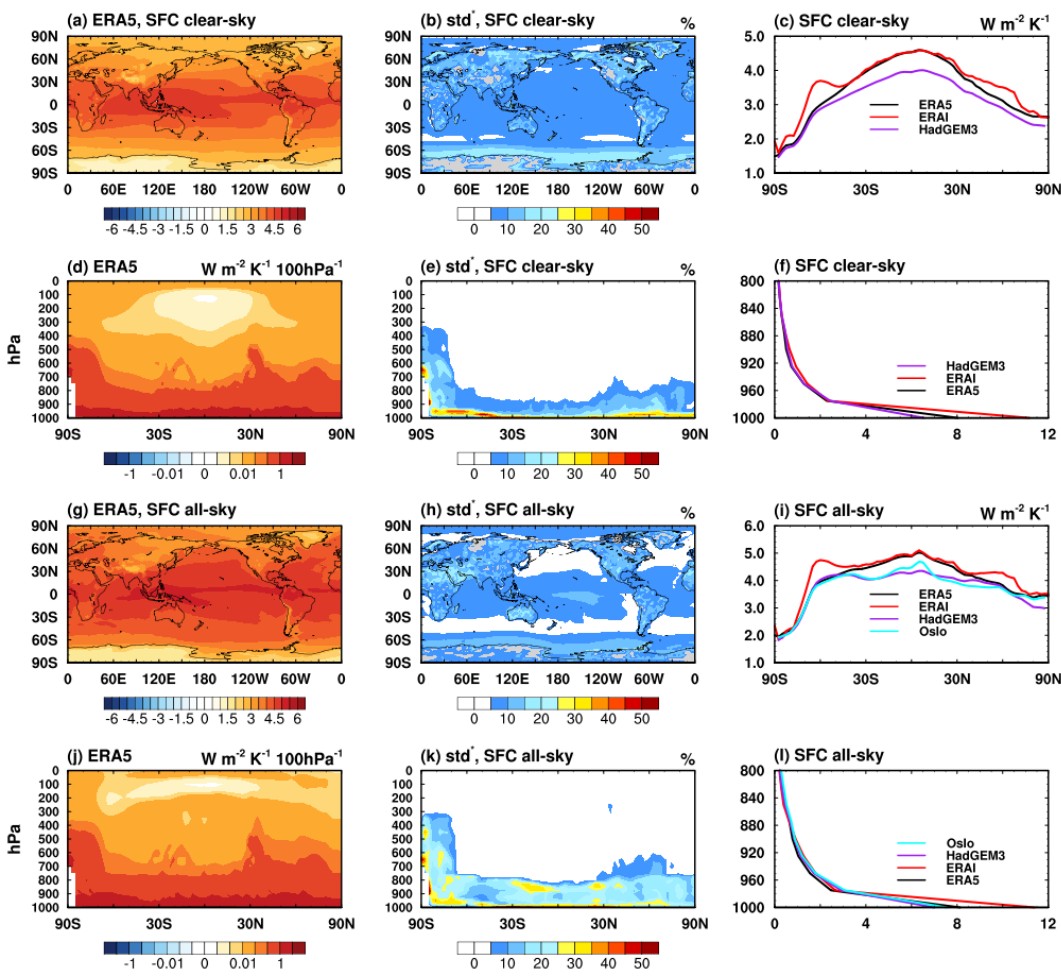

Figure 3. Like Figure 2, but for the sensitivity of surface flux to air temperature.

Figures 2 and 3 (left column) show the ERA5 air temperature kernels for TOA and surface
fluxes, respectively. The increase of air temperature increases the OLR at TOA and also the
downwelling flux at surface, so the TOA and surface kernels take negative and positive signs,
respectively. The TOA kernel has maximum values in the tropics, due to the higher air
temperature (Planck function) and more abundant cloud and water vapor (higher emissivity)
there, and generally decreases in magnitude with latitude. Unlike the TOA flux kernel (Figure
2d, j), which shows comparable sensitivity to air temperature at nearly all vertical levels, the
surface flux is mainly sensitive to the bottom layers (Figure 3d, j).



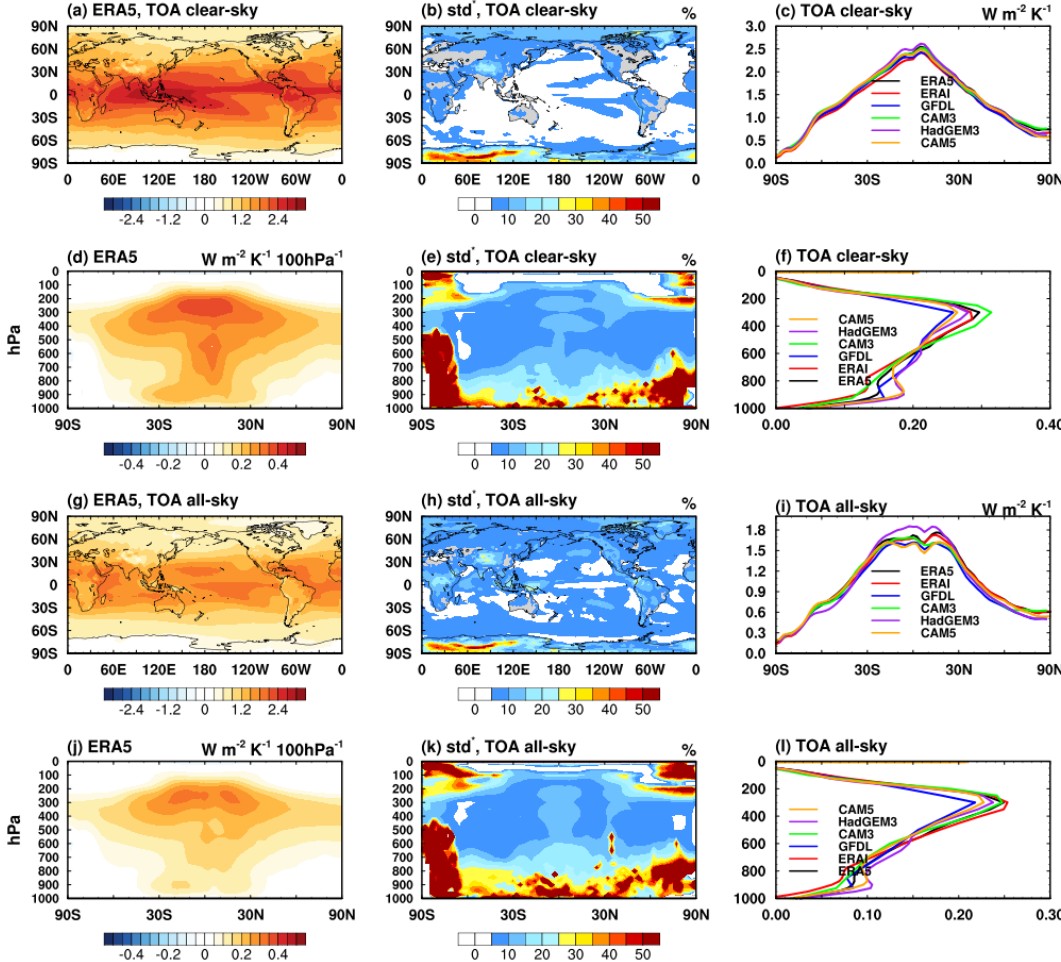

Figure 4. The sensitivity of the TOA LW flux to water vapor, in (a-f) clear-sky and (g-l)
all-sky. (a, d, g, j) The annual mean ERA5 kernels, among which (a) and (g) are the vertically
integrated sensitivity, units: W m$^{-2}$ K$^{-1}$, and (d) and (j) are the vertically resolved sensitivity,
units: W m$^{-2}$ K$^{-1}$ 100hPa$^{-1}$; (b, e, h, k) the fractional discrepancies of the radiative kernels listed
in Table 1; (c, i) zonal mean vertically integrated radiative kernels, units: W m$^{-2}$ K$^{-1}$; (g, l) global
mean vertically resolved kernels, units: W m$^{-2}$ K$^{-1}$ 100hPa$^{-1}$.

Earth System
Science
Data

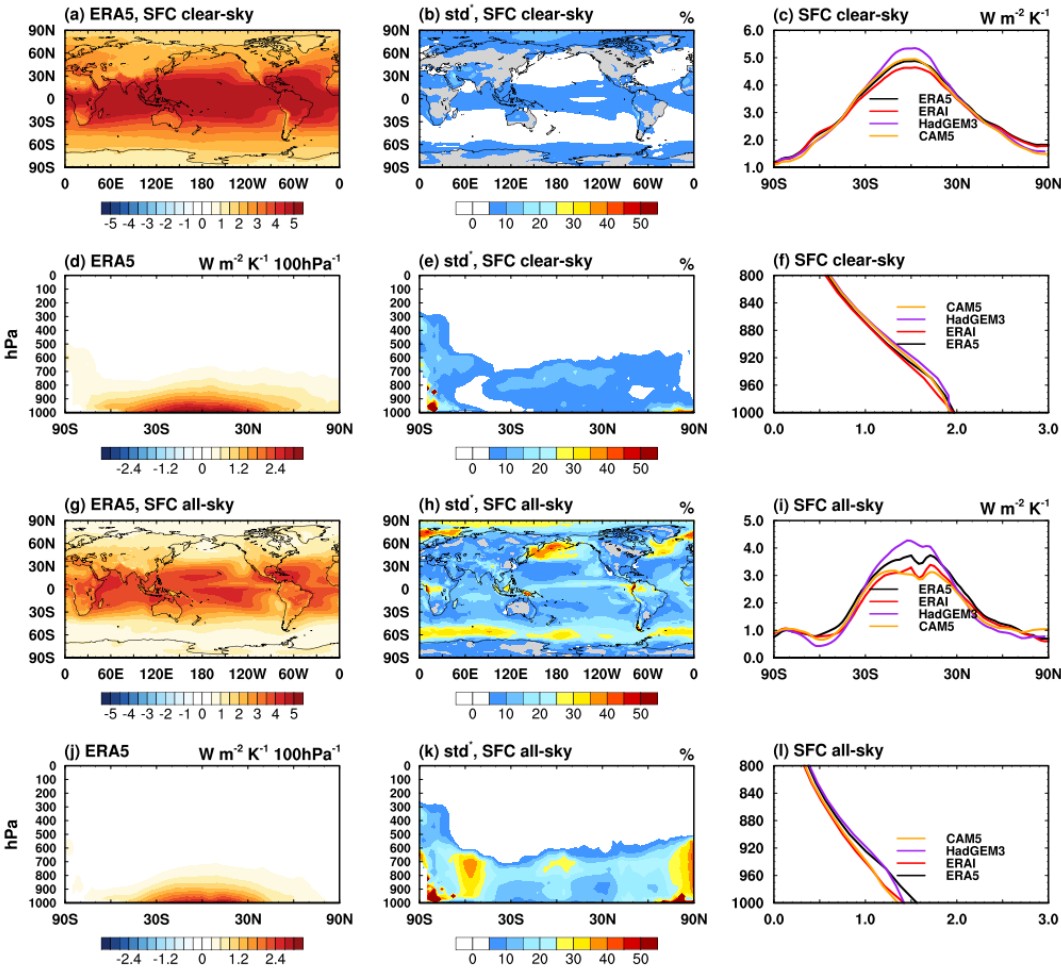

Figure 5. Like Figure 4, but for the sensitivity of the surface LW flux to water vapor.

Figures 4 and 5 (left column) show the ERA5 water vapor LW kernels for TOA and
surface fluxes, respectively. An increase of water vapor reduces OLR at TOA and increases
downwelling radiation at surface, so that the TOA and surface kernels are both positive in sign.
The vertically integrated kernel values (Figure 4a, g and 5a, g) generally follow the temperature
distribution, for example, decreasing in magnitude with latitude. In both cases, the kernel
magnitude is dampened by clouds in all-sky. The vertically resolved kernels show maximum
sensitivity of TOA flux to the upper troposphere (Figure 4d, j) and maximum sensitivity of
surface flux to the bottom layers (Figure 5d, j), respectively. Such features were discussed in
detail in previous works (e.g., Huang et al. 2007).



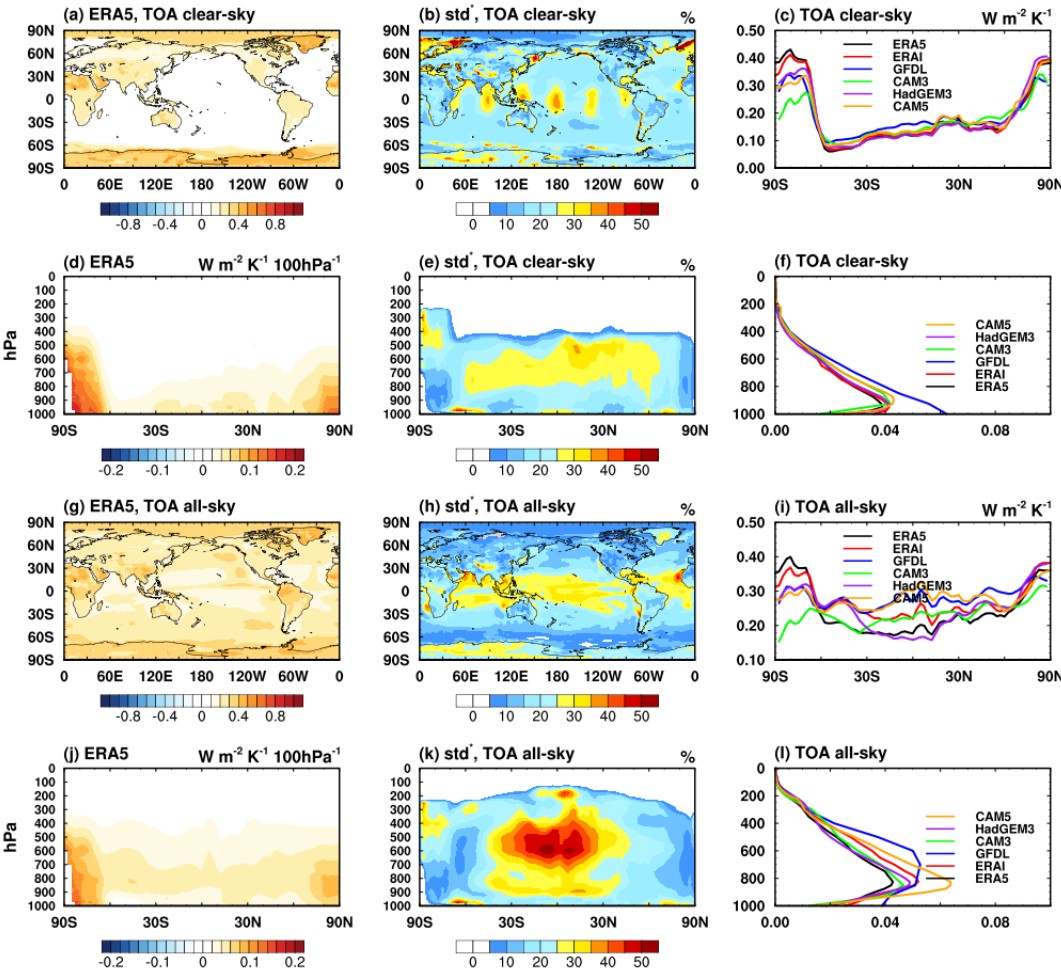

Figure 6. The sensitivity of the TOA SW flux to water vapor, in (a-f) clear-sky and (g-l) all-sky. (a, d, g, j) The annual mean ERA5 kernels, among which (a) and (g) are the vertically integrated sensitivity, units: W m$^{-2}$ K$^{-1}$, and (d) and (j) are the vertically resolved sensitivity, units: W m$^{-2}$ K$^{-1}$ 100hPa$^{-1}$; (b, e, h, k) the fractional discrepancies of the radiative kernels listed in Table 1; (c, i) zonal mean vertically integrated radiative kernels, units: W m$^{-2}$ K$^{-1}$; (g, l) global mean vertically resolved kernels, units: W m$^{-2}$ K$^{-1}$ 100hPa$^{-1}$.

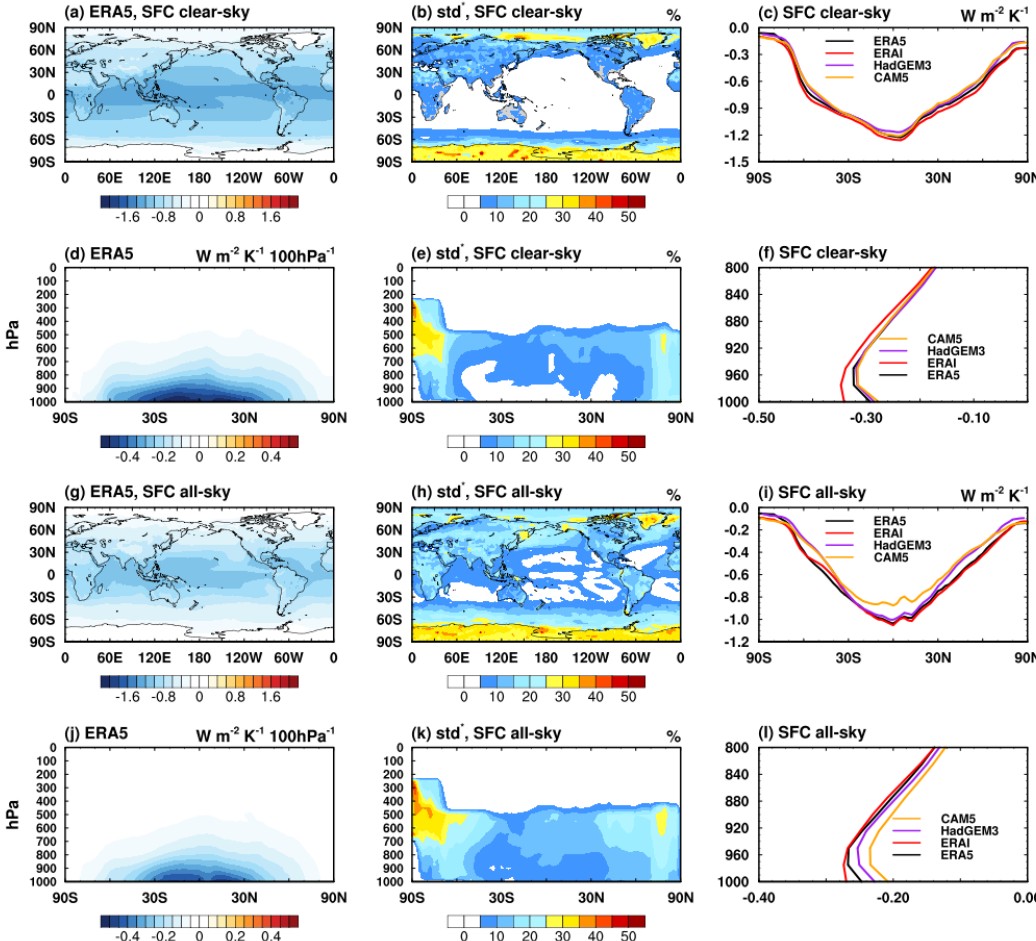

Figure 7. Like Figure 6, but for the sensitivity of the surface SW flux to water vapor.
Figures 6 and 7 (left column) show the ERA5 water vapor SW kernels for TOA and
surface fluxes, respectively. An increase of water vapor absorbs solar radiation and thus reduce
both the upwelling (reflected) SW flux at TOA and the downwelling SW flux at surface. As a
result, the two kernels take positive and negative signs, respectively. Note the magnitude of the
SW kernels is much weaker than that of the LW kernels, because water vapor absorbs the LW
flux more significantly than the SW flux. One noticeable feature of the TOA kernel (Figure 6a)
is that the magnitude over the land is stronger than that over the ocean, because the relatively
higher albedo over the land reflects more SW radiation and thus enhances the absorption by the
water vapor in the atmosphere. For this reason, over reflective surfaces such as the Sahara Desert
and Tibetan Plateau, as well as the Poles, the sensitivity is maximized. Unlike the TOA kernel,
the distribution of surface kernel follows the distribution of background water vapor
concentration, with noticeable dampening by clouds (Figure 7a, g).



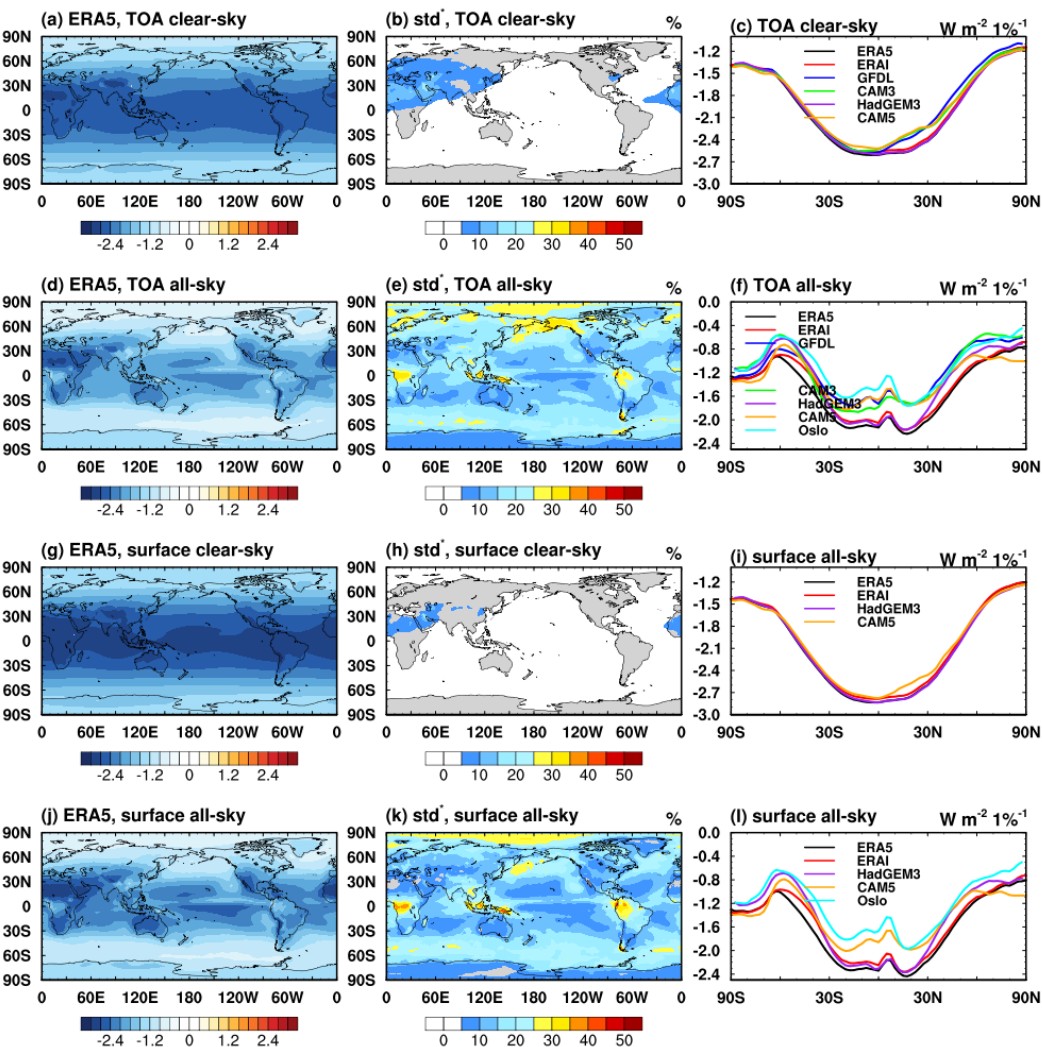

Figure 8. The sensitivity of (a-f) TOA and (g-l) surface fluxes to surface albedo, units: W
m$^{-2}$ 1%$^{-1}$. (a, d, g, j) The annual mean ERA5 sensitivity kernel; (b, e, h, k) the fractional
discrepancies of the kernels listed in Table 1; (c, f, i, l) zonal mean of the respective kernels.
Figure 8 (left column) illustrates the ERA5 surface albedo kernels, i.e., the sensitivity of
TOA and surface fluxes to surface albedo. An increase of surface albedo leads to more upwelling
(reflected) SW flux both at surface and TOA; therefore, the kernel is of negative sign. In clear-
sky, the sensitivity strength follows the pattern of solar insolation, with some local maxima, e.g.,
in the Sahara Desert and Tibetan Plateau (Figure 8a, g) due to the relatively lower water vapor
concentration. In all-sky, the distribution is again influenced by cloud patterns; for example, in
the ITCZ region, the strength is much reduced as clouds reduce the solar radiation reaching the
surface and thus the sensitivity to surface albedo change.
**3.2 Comparison of ERA5 kernels with other datasets**
To examine the discrepancies between different kernel datasets, we select six previously
published ones for comparison. Table 1 summarizes their resolutions and the atmospheric
datasets based on which they are computed, including the GCMs: GFDL (Soden et al., 2008),
CAM3 (Shell et al., 2008), CAM5 (Pendergrass et al., 2018), Oslo (Myhre et al., 2018) and
HadGEM3 (Smith et al., 2020) and also a global reanalysis: ERAi (Huang et al., 2017). This list
is meant to be representative instead of exhaustive.
Table 1. Summary of radiative kernels compared in this work. Datasets with * only have
TOA kernels. Oslo kernels are only available for all-sky.

| Radiative kernels | Horizontal resolution (lat*lon) | Vertical resolution | Reference |
|---|---|---|---|
| GFDL* | 2x2.5 | 17 (pressure level) | Soden et al., 2008 |
| CAM3* | 2.8x2.8 | 17 (pressure level) | Shell et al., 2008 |
| ERAi | 2.5x2.5 | 24 (pressure level) | Huang et al., 2017 |
| CAM5 | 0.94x1.25 | 30 (hybrid level) or 17 (pressure level) | Pendergrass et al., 2018 |
| Oslo | 2.8x2.8 | 60 (hybrid level) | Myhre et al., 2018 |
| HadGEM3 | 1.25x1.9 | 85 (hybrid level) or 19 (pressure level) | Smith et al., 2020 |
| ERA5 | 2.5x2.5 | 37 (pressure level) | This study |

To facilitate an intercomparison, these kernel datasets are interpolated to the same
horizontal and vertical resolutions as those of the ERA5 kernel when illustrated in the middle
and right columns in Figures 1 to 8. Note that the CAM5 and HadGEM3 kernels have two
versions, with one defined at the raw hybrid levels and the other interpolated to pressure levels.
To retain the accuracy of them as much as possible, the hybrid level version is used for the
interpolation and comparison in Figures 1-8, while in Section 4, the pressure level version is
used for quantifying the feedbacks of CMIP6 models. For the Oslo kernel, only surface
temperature, air temperature and surface albedo kernels in all-sky are available and hence it is
excluded for clear-sky comparisons. The GFDL and CAM3 kernels are only available for TOA
fluxes and are excluded for surface kernel comparisons.
Here we use the standard deviation ($std$) and its normalized value ($std^*$) to measure the
spread of the inter-kernel dataset biases:
$$std_X = \sqrt{\frac{1}{n-1}\sum_{i=1}^{n}\left(K_X^i - \overline{K_X}\right)^2} \quad (2)$$

$$std_X^* = \frac{std_X}{\overline{K_X}} * 100 \quad (3)$$

where $n$ is the total number of kernel datasets. $K_X^i$ is radiative kernel of variable $X$ from the $i^{th}$
dataset. $\overline{K_X}$ is the multi-dataset mean of radiative kernel $K_X$. The distributions of fractional
discrepancy ($std^*$) are shown in the middle columns of Figures 1 to 8. Note that some kernels



exhibit abnormal values, such as the surface and air temperature kernel of the surface flux in the
CAM5 kernel (see Appendix Figure A2) and the air temperature kernel of the TOA flux in the
Oslo kernel near the tropopause region (see Appendix Figure A4), indicating inconsistent
computation of their values, and thus are excluded in the corresponding $std_X^*$ statistics in Figures
1 to 8. See more discussions in Appendix.
The comparisons identify the following relatively larger biases in kernel values. Among
the TOA kernels, the surface temperature and albedo kernels show relatively large discrepancies
in the Arctic, Southern Ocean and over some continental regions in the tropics in all-sky (Figure
1e and 8e), with the maximum discrepancy exceeding 30%; the air temperature kernel shows
larger discrepancies in the lower troposphere and tropical tropopause region (Figure 2k); these
kernel biases are likely due to the differences in cloud fields. The water vapor LW kernel also
shows noticeable fractional biases, for example, over the Antarctic region (Figure 4b and h). The
water vapor SW kernel shows biases in the tropical mid-troposphere and over Antarctic in both
clear-sky and all-sky (Figure 6e and 6k), leading to strong variations in the vertical integration of
sensitivity (Figure 6b and 6h), with a spread exceeding 30%. The noticeable periodic equatorial
pattern in Figure 6b is caused by the CAM3 kernel, likely due to a coarser temporal resolution
that does not well resolve the diurnal cycle of solar insolation in the kernel computation.
For the surface kernels, the most prominent biases exist in SW radiative kernels (Figure 7
and 8), especially in the polar regions. The discrepancy in the water vapor SW kernel reaches
30% for vertically integrated values (Figure 7b and h), with noticeable biases through the
troposphere (Figure 7e and k). The surface albedo kernel biases are much larger in all-sky than
that in clear-sky, indicating that the cause is in cloud fields, and are also noticeable in the Arctic
region due to sea ice variations (Figure 8). In the LW, the water vapor kernels exhibit noticeable
biases in the Central Pacific, Southern Ocean and Arctic in all-sky, where again the difference in
cloud field is likely the cause. The air temperature kernels show noticeable discrepancies in the
bottom layers (Figure 3e), which may be caused by the inconsistency in the kernel computation
(see the discussions in Appendix).
In summary, the biases among radiative kernel datasets are generally smaller in clear-sky
than in all-sky and in most cases, and are largely within 10%. However, there are some notable
regional biases, for example, in the surface temperature kernel in the tropics (Figure 1e), in the
surface albedo kernel in the Arctic (Figure 8e and k), and in the water vapor SW kernel in the
Antarctic region (Figure 7b and h), likely due to the dependence of radiative sensitivity on
background climate states, which differ between the kernel datasets. To ascertain the state-
dependency caused kernel uncertainty, we next examine the ERA5 kernels computed from
different years.
**3.3 Interannual variation of kernel values**
The intercomparison above identified several prominent inter-dataset biases in the kernel
values. For example, there are noticeable differences in the values of surface temperature, albedo
and water vapor kernels in the Central Pacific and Arctic region. One possible reason that may
account for such differences is the atmospheric state-dependency of the kernel values. Besides
the inter-model biases in the different GCM climatology, the interannual variations of the
atmospheric states, such as the cloudiness variations in the Central Pacific region during the
ENSO cycle, may affect the radiative sensitivity. To test this hypothesis, we use the ENSO and



sea ice loss cases to demonstrate the changes in radiative sensitivity with a focus on Central
Pacific and Arctic region, respectively. In the ENSO case, the variation is defined as the
difference in annual mean kernel values between 2011 and 2015, which have the seasonal sea
surface temperature anomalies in the Niño 3.4 region (5N-5S, 190-240E) of -1.4K and +2.6K,
respectively. In the sea ice loss case, the variation is calculated as the difference in September
between year 2012 and 2013, as the sea ice cover in 2012 was reported to be the lowest level
since satellite observation era.
364   To save space, here we only exemplify the most prominent differences. Figure 9
summarizes the differences in all-sky for both TOA and surface fluxes due to ENSO. The
increases of water vapor concentration and cloudiness in the Central Pacific greatly dampen the
sensitivity of OLR to surface temperature change (Figure 9d), with maximum reduction reaching
60% (Figure 9d). Also noticed is a weakening of the surface LW sensitivity to water vapor
(Figure 9f) and an increase of the TOA SW sensitivity to water vapor (Figure 9e), due to the
effects of clouds.

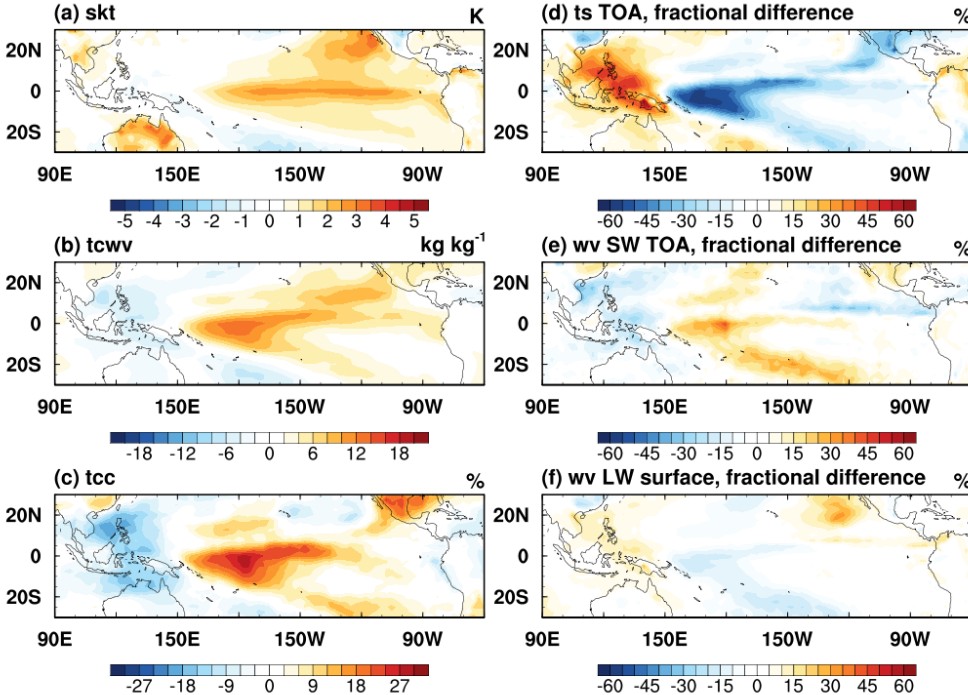

373   Figure 9. Annual mean differences between year 2015 and 2011 in (a) surface skin
temperature, (b) total column water vapor, (c) total cloud cover, and the fractional differences in
(d) surface temperature kernel for TOA LW flux, (e) water vapor kernel for TOA SW flux, (f)
water vapor kernel for surface LW flux. The fractional difference is quantified relative to multi-
year mean.

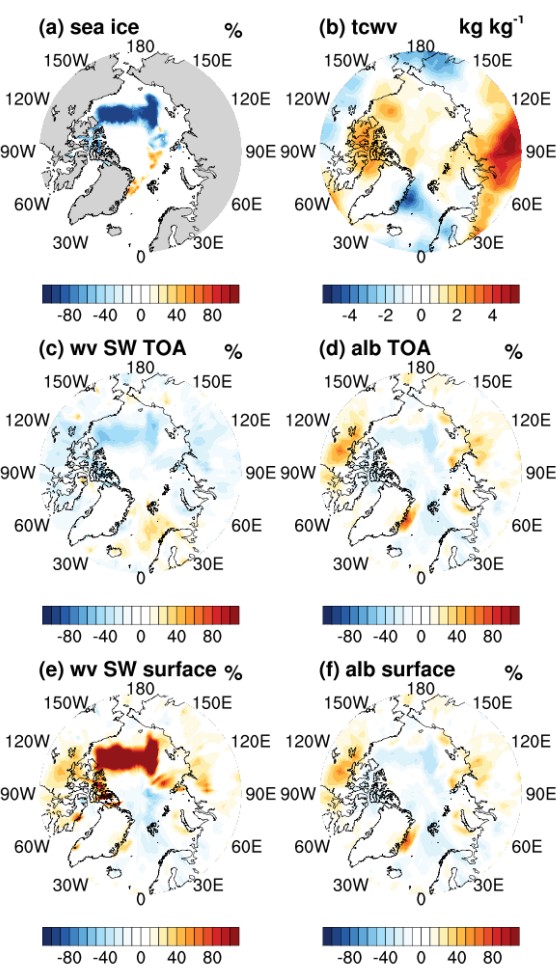

Figure 10. September differences between 2012 and 2013 in (a) sea ice concentration, (b)
total column water vapor, and the fractional differences in (c, e) water vapor SW kernel for TOA
and surface fluxes, (d, f) surface albedo kernel for TOA and surface fluxes. The fractional
difference is measured relative to values in September 2013.
In the sea ice loss case, the reduction of sea ice in the Arctic region (Figure 10a) leads to a
significant decrease of radiative sensitivity to surface albedo (Figure 10d and f), with the
maximum difference exceeding 50%, as well as a significant decrease in the TOA sensitivity and
an increase of surface sensitivity to water vapor, respectively (Figure 10c and e), with the
maximum changes exceeding 80%.
In summary, these quantitatively large interannual differences, as well as their locations,
which correspond to the inter-dataset kernel differences seen in Figure 1-8, affirm that some
discrepancies between the radiative kernels are caused by the difference in atmospheric states.
Nevertheless, it ought to be noted that the differences are localized and because of that do not
cause significant biases in the global mean feedback values.

**4. Feedback quantification**
In this section, we apply different kernels to quantifying the radiative feedbacks in one
quadrupling $CO_2$ experiment (abrupt4x$CO_2$) of CMIP6 models. This experiment is selected
because it has been used by a number of studies for forcing and feedback analyses (e.g., Zelinka
et al., 2020), which we can compare our results to. The CMIP6 models used in this assessment
are listed in Table2. Note that the standard outputs at 19 pressure levels from the models and
correspondingly the kernel values, including CAM5 and HadGEM3, provided at the pressure
levels are used in this section.
Table 2. Summary of CMIP6 models used in this study.

| Models | Horizontal resolution (lat*lon) | Model top level (hPa) | Reference |
|---|---|---|---|
| CESM2 | 0.9*1.25 | 32 levels to 2.26 hPa | Danabasoglu et al. (2020) |
| CNRM-CM6-1 | 1.4*1.4 | 91 levels to 0.01hPa | Voldoire et al. (2019) |
| EC-Earth3 | 0.7*0.7 | 91 levels to 90 km | Döscher et al. (2022) |
| HadGEM3-GC31-LL | 1.25*1.875 | 85 levels to 85km | Williams et al. (2018) |
| IPSL-CM6A-LR | 1.3*2.5 | 79 levels to 80km | Boucher et al. (2020) |
| MPI-ESM1-2-LR | 1.875*1.875 | 47 levels to 0.01hPa | Mauritsen et al. (2019) |

**4.1 Analysis procedure**
To quantify the radiative feedbacks, data from two experiments as documented by Eyring
et al. (2016) are used: abrupt4x$CO_2$, simulations with an instantaneous quadrupling of $CO_2$
concentration of year 1850, piClim-4x$CO_2$, simulations with SST and sea ice concentrations
fixed at the climatology of pre-industrial control experiment and $CO_2$ concentration quadrupled.
In each experiment, a 20-year period at the end of the simulation in each model is used.
Following the previous studies (e.g., Smith et al., 2020; Zelinka et al., 2020), radiative feedbacks
are diagnosed using the difference of atmospheric variables between the abrupt4x$CO_2$ and
piClim-4x$CO_2$ experiments.
To detail the analysis procedure, firstly, all variables including radiative fluxes and
atmospheric variables from CMIP6 models are interpolated to the horizontal and vertical
resolution of the kernel itself. Notre that for CAM3, GFDL and CAM5 kernels, they only have
17 pressure level which are two layers (1hPa and 5hPa) fewer than the CMIP6 standard model
output. To address this issue, the contribution of the two missing layers is calculated using other
kernels (e.g., ERA5) and found to have negligible effect on the global mean feedback value.
Hence, when using these three kernels, the contributions from 10hPa above are ignored.
Secondly, the non-cloud radiative feedback of atmospheric variable $X$ is calculated as:
$$\Delta R_X = K_X \cdot \Delta X \quad (4)$$
with units in W m$^{-2}$. For the 2D radiative kernels (surface temperature and surface albedo), $K_X$
and $\Delta X$ have just single layer values and $\Delta R_X$ is simply the product of these two terms. For the
3D radiative kernels (air temperature and water vapor), both $K_X$ and $\Delta X$ are vectors of pressure

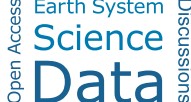

levels and $\Delta R_X$ is the dot product of $K_X$ and $\Delta X$. Note that if $K_X$ is normalized with unit pressure
thickness (e.g., W m$^{-2}$ K$^{-1}$ 100hPa$^{-1}$), the layer thickness must be taken into account when
calculating $dR_X$.
Finally, cloud feedbacks are diagnosed using the adjusted cloud-radiative forcing method
(Shell et all., 2008). Here we compute the residual term in clear-sky as:
$res^o = \sum \Delta R_X^o - \Delta R^o$          (5)
and, assuming the all-sky decomposition has the same non-closure residual, the cloud feedback
is measured as
$\Delta R_c = \Delta R - \sum \Delta R_X + res^o$          (6)
where the superscript $o$ represents clear-sky quantities. $\sum \Delta R_X^o$ and $\sum \Delta R_X$ are the sum of non-
cloud feedbacks in clear-sky and all-sky, respectively, diagnosed by multiplying the radiative
kernel with the atmospheric responses measured as the difference between abrupt4xCO$_2$ and
piClim-4xCO$_2$ experiments. $\Delta R^o$ and $\Delta R$ are the total radiation change in clear-sky and all-sky,
respectively, calculated as the difference in the GCM-simulated radiative fluxes between two
experiments.
The feedback parameters, $\lambda_X$, in the units of W m$^{-2}$ K$^{-1}$, are then obtained by normalizing
the feedback flux changes $\Delta R_X$ by the global mean surface temperature change $\Delta T_S$ in the
abrupt4xCO$_2$ experiment:
$\lambda_X = \Delta R_X / \Delta T_S$          (7)

**4.2 TOA feedbacks**

The residual term ($res^o$) measures the unexplained radiation change in the feedback
analysis and provides a useful overall indication of the soundness of the feedback quantification.
Figure 11 illustrates the residual term for the TOA flux decomposition when different kernels are
used to diagnose the feedback of the HadGEM3-GC3-LL model. Note that the findings described
below are insensitive to which GCM is used. In terms of the global mean, all residual terms are
of small magnitude, no matter which kernel dataset is used. However, there are some noticeable
local residuals, especially for the SW budget, e.g., in the Arctic region and around the Antarctic
continent where sea ice changes the most (mid-column in Figure 11). While the non-zero
magnitude of the residual is partly due to nonlinearity in the radiation decomposition, e.g.,
possible coupling between surface albedo and water vapor (Y. Huang et al., 2021), the spread
among the kernel results as evidenced by the line plots of Figure 11 is attributable to the
discrepancies in the SW radiative kernels as revealed by the comparisons in Section 3. In the
LW, the residual is generally small. In summary, the residual terms for the TOA budget are small
in terms of the global mean feedback strengths, affirming the validity of the radiative kernels for
feedback quantification.
Table 3 summarizes the total feedback parameters quantified by normalizing the total
radiative feedback in all-sky (the $\sum \Delta R_X$ term in Equation 6) with the global mean surface
temperature change in each model. These results are consistent with other published results. For
example, compared with the results of Zelinka et al. (2020) based on the ERAi kernel, the
kernel-diagnosed overall feedback parameter in the two results is -0.87 W m$^{-2}$ K$^{-1}$ and -0.85 W
m$^{-2}$ K$^{-1}$ for the CNRM-CM6-1 model and -0.81 W m$^{-2}$ K$^{-1}$ and -0.84 W m$^{-2}$ K$^{-1}$ for the
HadGEM3-GC3-LL model. The results in Table 3 show that although there are larger local
discrepancies, the total feedback strengths diagnosed from different kernels (the 9[th] rows in





Table 3) are in good agreement, with a spread generally within 10%. In comparison, the
discrepancy in the overall feedback among different models, even when analyzed by the same
kernel dataset (the last column in Table 3), is much larger, amounting to 20% for LW, 70% for
SW and 30% for the net. In general, the overall feedback uncertainty as measured by inter-model
spread primarily arises from the different climate responses simulated in these models rather than
the kernel uncertainty.
Table 4 summaries the individual TOA feedback components. For the non-cloud
feedbacks in each model analyzed from different kernel datasets (numbers in each column in
Table 4), the spreads of LW feedback components are generally within 5%; the spreads in the
SW feedback components are within 20%. For the cloud feedbacks, the spreads caused by
different kernels are larger. For example, the spread of the LW cloud feedback is 36% in the
CESM2 model. The spreads of the SW cloud feedback can exceed 100% in some models. These
numbers are consistent with the kernel-induced feedback differences assessed by others (e.g.,
Vial et al. 2013). However, it is important to note that these spreads caused by the kernel
difference are still less than the inter-GCM spreads of the cloud feedbacks (see the last column of
Table 4).
In summary, the inter-kernel biases do not lead to significant uncertainty in the analyzed
non-cloud feedbacks; although the kernel-induced uncertainty in cloud feedback is larger, this
uncertainty is considerably less than the inter-GCM cloud feedback spread.





Table 3. Kernel-diagnosed all-sky TOA overall feedback parameter of CMIP6 models, units: W m⁻² K⁻¹. The numbers in the brackets of the ERAi kernel row are the results from Zelinka et al. (2020) for comparison. The bolded rows: the mean and standard deviation of the feedback parameter values diagnosed from different kernels for the same GCM; the last column: the mean and standard deviation of the feedback parameter values of different GCMs diagnosed from a same kernel, units: W m⁻² K⁻¹.

| | CESM2 | | | CNRM-CM6-1 | | | EC-Earth3 | | | HadGEM3-GC3-LL | | | IPSL-CM6A-LR | | | MPI-ESM1-2-LR | | | Multi-model mean | | |
|---|---|---|---|---|---|---|---|---|---|---|---|---|---|---|---|---|---|---|---|---|---|
| | LW | SW | Net | LW | SW | Net | LW | SW | Net | LW | SW | Net | LW | SW | Net | LW | SW | Net | LW | SW | Net |
| ERA5 | -1.95 | 1.28 | -0.67 | -1.40 | 0.62 | -0.78 | -1.55 | 0.77 | -0.78 | -2.36 | 1.59 | -0.78 | -1.48 | 0.63 | -0.85 | -1.41 | -0.02 | -1.43 | -1.69±0.39 | 0.81±0.56 | -0.88±0.28 |
| ERAi | -2.01 | 1.27 | -0.74 (-0.64) | -1.46 | 0.61 | -0.85 (-0.87) | -1.61 | 0.76 | -0.85 (-0.89) | -2.42 | 1.57 | -0.84 (-0.81) | -1.55 | 0.62 | -0.93 (-0.97) | -1.48 | -0.03 | -1.51 | -1.75±0.38 | 0.80±0.56 | -0.95±0.28 |
| CAM3 | -1.88 | 1.28 | -0.60 | -1.32 | 0.62 | -0.70 | -1.50 | 0.77 | -0.72 | -2.31 | 1.59 | -0.72 | -1.42 | 0.63 | -0.79 | -1.35 | -0.01 | -1.36 | -1.63±0.39 | 0.81±0.56 | -0.81±0.27 |
| GFDL | -2.04 | 1.28 | -0.76 | -1.50 | 0.61 | -0.90 | -1.65 | 0.75 | -0.90 | -2.46 | 1.57 | -0.89 | -1.60 | 0.62 | -0.98 | -1.55 | -0.02 | -1.57 | -1.80±0.37 | 0.80±0.56 | -1.00±0.29 |
| CAM5 | -1.97 | 1.30 | -0.67 | -1.42 | 0.64 | -0.78 | -1.58 | 0.79 | -0.78 | -2.38 | 1.61 | -0.78 | -1.51 | 0.64 | -0.86 | -1.44 | 0.00 | -1.43 | -1.72±0.38 | 0.83±0.56 | -0.88±0.28 |
| HadGEM3 | -1.92 | 1.26 | -0.65 | -1.39 | 0.61 | -0.65 | -1.55 | 0.77 | -0.78 | -2.35 | 1.58 | -0.77 | -1.47 | 0.62 | -0.85 | -1.40 | -0.03 | -1.43 | -1.68±0.38 | 0.80±0.56 | -0.88±0.28 |
| **Multi-kernel mean** | **-1.96 ±0.06** | **1.28 ±0.01** | **-0.68±0.06** | **-1.42 ± 0.06** | **0.62 ±0.01** | **-0.80 ±0.07** | **-1.57 ±0.05** | **0.77 ±0.01** | **-0.80 ±0.06** | **-2.38 ±0.05** | **1.58 ±0.01** | **-0.79 ±0.06** | **-1.50 ±0.06** | **0.63 ±0.01** | **-0.88 ±0.07** | **-1.44 ±0.07** | **-0.02 ±0.01** | **-1.45 ±0.07** | | | |
| **Multi-kernel $\lambda_{res}$** | **-0.05 ±0.06** | **0.15 ±0.01** | **0.11±0.06** | **0.03 ±0.06** | **0.04 ±0.01** | **0.07 ±0.07** | **0.07 ±0.05** | **0.08 ±0.01** | **0.15 ±0.06** | **-0.06 ±0.05** | **0.04 ±0.01** | **0.02 ±0.06** | **0.00 ±0.06** | **0.03 ±0.01** | **0.03 ±0.07** | **0.09 ±0.07** | **0.04 ±0.01** | **0.13 ±0.07** | | | |
| $dT_s$ (K) | 11.16 | | | 6.59 | | | 7.18 | | | 7.61 | | | 8.24 | | | 4.70 | | | | | |






Table 4. Kernel-diagnosed all-sky TOA radiative feedback parameters, units: W m$^{-2}$ K$^{-1}$.
First six columns: the mean and standard deviation of the feedback parameter values diagnosed
from different kernels for the same GCM; the last column: the mean and standard deviation of
the feedback parameter values of different GCMs diagnosed from multi-kernel mean results,
units: W m$^{-2}$ K$^{-1}$.

|  | CESM2 | CNRM-CM6-1 | EC-Earth3 | HadGEM3-GC3-LL | IPSL-CM6A-LR | MPI-ESM1-2-LR | **Multi-model** |
|---|---|---|---|---|---|---|---|
| $\lambda_{Ts}$ | -0.68±0.02 | -0.65±0.03 | -0.64±0.03 | -0.66±0.03 | -0.63±0.03 | -0.63±0.03 | **-0.65±0.02** |
| $\lambda_{Ta}$ | -3.01±0.05 | -2.92±0.04 | -2.77±0.04 | -3.06±0.04 | -2.83±0.04 | -3.13±0.05 | **-2.95±0.14** |
| $\lambda_q$ LW | 1.60±0.05 | 1.58±0.06 | 1.54±0.05 | 1.51±0.05 | 1.64±0.06 | 1.66±0.06 | **1.59±0.06** |
| $\lambda_c$ LW | 0.13±0.05 | 0.58±0.05 | 0.30±0.04 | -0.17±0.04 | 0.31±0.05 | 0.67±0.05 | **0.30±0.30** |
| $\lambda_q$ SW | 0.25±0.03 | 0.24±0.03 | 0.26±0.03 | 0.24±0.03 | 0.25±0.03 | 0.26±0.03 | **0.25±0.01** |
| $\lambda_{alb}$ | 0.28±0.05 | 0.40±0.07 | 0.44±0.08 | 0.38±0.06 | 0.33±0.05 | 0.32±0.06 | **0.36±0.06** |
| $\lambda_c$ SW | 0.74±0.06 | -0.02±0.07 | 0.07±0.08 | 0.97±0.07 | 0.04±0.06 | -0.59±0.07 | **0.20±0.57** |






Table 5. Similar to Table 3, but for the surface feedback. The overall feedback parameter and residual term from HadGEM3 kernel are shown in last two rows for comparison.

| | CESM2 | | | CNRM-CM6-1 | | | EC-Earth3 | | | HadGEM3-GC3-LL | | | IPSL-CM6A-LR | | | MPI-ESM1-2-LR | | | Multi-model mean | | |
|---|---|---|---|---|---|---|---|---|---|---|---|---|---|---|---|---|---|---|---|---|---|
| | LW | SW | Net | LW | SW | Net | LW | SW | Net | LW | SW | Net | LW | SW | Net | LW | SW | Net | LW | SW | Net |
| ERA5 | 0.64 | 0.55 | 1.19 | 0.85 | -0.17 | 0.68 | 1.02 | -0.01 | 1.00 | 0.67 | 0.80 | 1.48 | 0.85 | -0.20 | 0.65 | 1.29 | -0.87 | 0.43 | 0.89±0.24 | 0.02±0.60 | 0.90±0.39 |
| ERAi | 0.56 | 0.52 | 1.08 | 0.79 | -0.20 | 0.59 | 0.95 | -0.05 | 0.90 | 0.57 | 0.77 | 1.35 | 0.74 | -0.23 | 0.51 | 1.11 | -0.90 | 0.21 | 0.79±0.22 | -0.01±0.60 | 0.77±0.41 |
| CAM5 | 0.53 | 0.60 | 1.12 | 0.69 | -0.12 | 0.57 | 0.99 | 0.03 | 1.02 | 0.58 | 0.85 | 1.43 | 0.76 | -0.15 | 0.61 | 1.27 | -0.83 | 0.44 | 0.80±0.28 | 0.06±0.60 | 0.87±0.39 |
| Multi-kernel mean | 0.57±0.06 | 0.56±0.04 | 1.13±0.06 | 0.78±0.08 | -0.17±0.04 | 0.61±0.06 | 0.99±0.03 | -0.01±0.04 | 0.98±0.07 | 0.61±0.06 | 0.81±0.04 | 1.42±0.07 | 0.78±0.06 | -0.19±0.04 | 0.59±0.07 | 1.22±0.10 | -0.87±0.04 | 0.36±0.13 | | | |
| Multi-kernel $\lambda_{res}$ | -0.25±0.06 | 0.22±0.04 | -0.03±0.06 | -0.14±0.08 | 0.05±0.04 | -0.19±0.06 | 0.00±0.03 | 0.08±0.04 | 0.08±0.06 | -0.11±0.06 | 0.16±0.04 | 0.05±0.07 | -0.10±0.06 | 0.03±0.04 | -0.13±0.07 | 0.13±0.10 | 0.04±0.03 | 0.18±0.13 | | | |
| HadGEM3 | -2.69 | 0.70 | -1.99 | -2.27 | -0.02 | -2.30 | -2.08 | 0.14 | -1.94 | -2.58 | 0.96 | -1.63 | -2.25 | -0.05 | -2.30 | -1.89 | -0.72 | -2.61 | | | |
| HadGEM3 $\lambda_{res}$ | -3.52 | 0.37 | -3.15 | -3.19 | 0.09 | -3.10 | -3.06 | 0.23 | -2.84 | -3.30 | 0.31 | -2.99 | -3.14 | 0.12 | -3.02 | -2.98 | 0.18 | -2.79 | | | |





Table 6. Similar to Table 4, but for the surface feedbacks. The temperature feedback
parameters from HadGEM3 kernel are shown in last three rows for comparison.

| | CESM2 | CNRM-CM6-1 | EC-Earth3 | HadGEM3-GC3-LL | IPSL-CM6A-LR | MPI-ESM1-2-LR | Multi-model mean |
|---|---|---|---|---|---|---|---|
| $\lambda_{Ta} + \lambda_{Ts}$ | -1.14±0.08 | -1.13±0.08 | -1.08±0.06 | -1.07±0.05 | -1.04±0.05 | -0.89±0.00 | **-1.06±0.09** |
| $\lambda_{Ts}$ | -4.00±1.00 | -3.82±0.96 | -3.80±0.96 | -3.96±1.00 | -3.72±0.93 | -3.75±0.95 | **-3.84±0.11** |
| $\lambda_{Ta}$ | 2.87±0.97 | 2.70±0.93 | 2.72±0.91 | 2.89±0.97 | 2.68±0.90 | 2.87±0.94 | **2.79±0.10** |
| $\lambda_q$ LW | 1.99±0.15 | 1.85±0.15 | 2.01±0.15 | 1.92±0.14 | 1.93±0.15 | 2.01±0.15 | **1.95±0.06** |
| $\lambda_c$ LW | -0.28±0.08 | 0.05±0.08 | 0.06±0.09 | -0.24±0.07 | -0.11±0.09 | 0.10±0.10 | **-0.07±0.16** |
| $\lambda_q$ SW | -0.63±0.05 | -0.59±0.06 | -0.63±0.06 | -0.59±0.05 | -0.61±0.06 | -0.64±0.05 | **-0.62±0.02** |
| $\lambda_{alb}$ | 0.34±0.02 | 0.49±0.02 | 0.54±0.02 | 0.46±0.02 | 0.40±0.01 | 0.38±0.03 | **0.44±0.07** |
| $\lambda_c$ SW | 0.84±0.04 | -0.07±0.04 | 0.07±0.04 | 0.95±0.04 | 0.02±0.04 | -0.61±0.05 | **0.20±0.59** |
| | | | | | | | |
| HadGEM3 $\lambda_{Ta} + \lambda_{Ts}$ | -3.86 | -3.70 | -3.66 | -3.76 | -3.60 | -3.54 | |
| HadGEM3 $\lambda_{Ts}$ | -5.12 | -4.89 | -4.86 | -5.06 | -4.74 | -4.79 | |
| HadGEM3 $\lambda_{Ta}$ | 1.26 | 1.19 | 1.20 | 1.29 | 1.15 | 1.26 | |


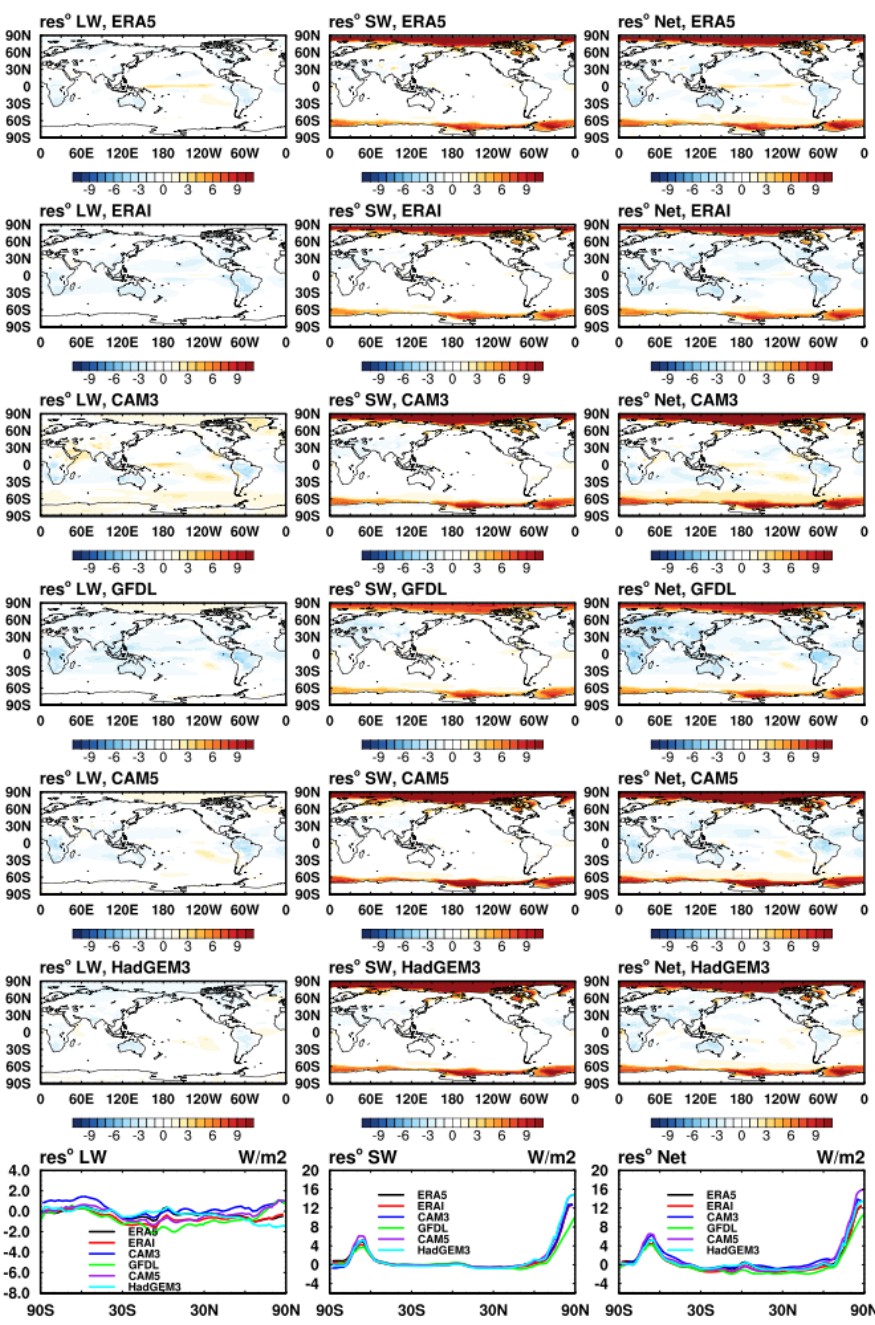

Figure 11. The TOA budget residual ($res^o$) in the HadGEM3-GC3-LL model when the
feedbacks are calculated by different kernels. (left column) LW, (mid-column) SW, (right-
column) Net, the sum of LW and SW. The three line-plots in the bottom row are the zonal mean
residuals.




**4.3 Surface feedbacks**
Next, we examine how the inter-kernel biases lead to uncertainty in the analyzed surface
feedbacks. Figure 12 shows the residual distribution and Table 5 and 6 summarize the global
mean feedback strengths.
First of all, we find that when the ERA5 and ERAi kernels are used for the feedback
analysis, the non-closure residual in the surface budget is comparable in magnitude to the TOA
analysis. This suggests that the surface kernels afford a valid tool for the surface feedback
analysis. However, some prominent biases are noticed for other kernel datasets. For example, the
HadGEM3 kernels, show especially an underestimation in air temperature feedback, likely due to
a biased sensitivity of the bottom atmospheric layer (see Appendix for more discussions). The
global mean air temperature feedback parameter measured by the HadGEM3 kernel is around
1.20 W m$^{-2}$ K$^{-1}$ (compared to around 2.80 W m$^{-2}$ K$^{-1}$ measured by the other kernels), and the
non-closure residual is as large as -3.0 W m$^{-2}$ K$^{-1}$ (compared to 0.1 W m$^{-2}$ K$^{-1}$ in the others). For
this reason, the result from HadGEM3 kernel is excluded for the multi-kernel statistics in Table
5, but listed in a separate row in Tables 5 and 6 for comparison. On the other hand, the results
here show that, similar to the TOA assessment, the overall feedback spread across the models
(last column in Table 5) is greater than the uncertainty caused by kernel biases.
Table 6 lists the individual surface feedback components. For the temperature feedbacks
and for the reasons discussed in Appendix, it is advisable to examine the sum of surface and air
temperature feedbacks, rather than their separate values as a measure of the inter-kernel biases.
For the non-cloud feedbacks, the inter-kernel bias-caused feedback uncertainty is generally
within 10%, It is noted that for some feedbacks, such as the water vapor LW and SW feedbacks,
the inter-kernel uncertainty exceeds the inter-model uncertainty, which is linked to the large
discrepancies detected in the radiative kernel values as shown by Figure 5h and 7h. For the cloud
feedbacks, although the kernel-induced uncertainty is higher, with the maximum exceeding
100% in some cases, this uncertainty is still less than the inter-model spread of the feedback.
In summary, we find the surface feedback decomposition can achieve similar level of
radiation closure to the TOA analysis, affirming the validity of kernels for diagnosing the surface
radiative feedback. However, the results qualitatively vary depending on which kernel dataset is
used, indicating errors in the computation of some kernels.



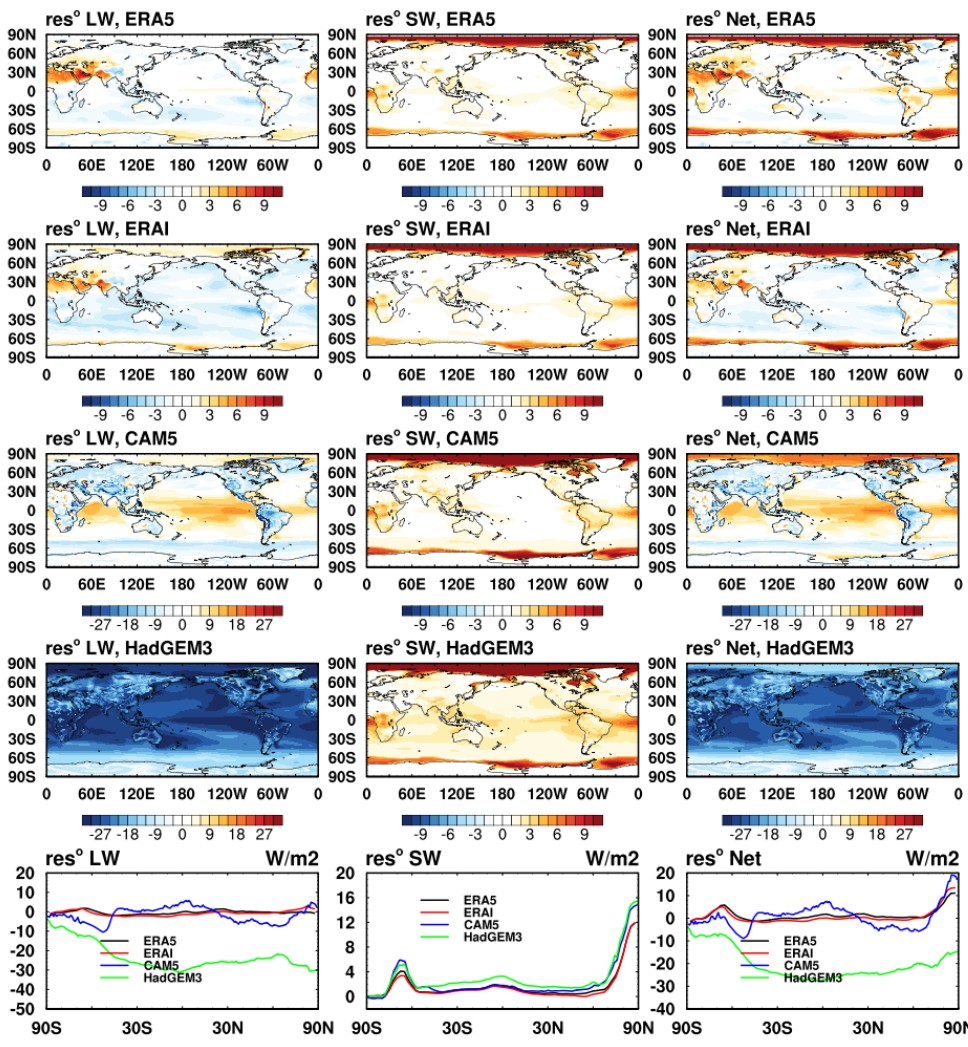

Figure 12. Similar to Figure 11, but for the surface feedback analysis.

## 5. Data availability

The datasets contain the multi-year averaged monthly mean TOA and surface kernel for surface temperature, air temperature, surface albedo and water vapor (LW and SW) and are available at: http://dx.doi.org/10.17632/vmg3s67568.1 (H.Huang, 2022).

## 6. Conclusions and discussions

In this paper, we present a newly generated set of ERA5-based radiative kernels of surface and air temperatures, water vapor and surface albedo, and compare them with other



published kernels, in terms of the kernel values, as well as the radiative feedbacks quantified
from them for both the TOA and surface radiation budgets.
For the TOA kernels, the results here affirm general consistency among the different
kernel datasets, and the discrepancies are generally within 10% in terms of vertically integrated
or globally averaged radiative sensitivity, although some relatively larger regional biases are
noticed, including those in the surface temperature kernel in the tropics (Figure 1), those in the
surface albedo kernel in the Arctic (Figure 8) and those in the water vapor shortwave kernel in
the Antarctica (Figure 6), which is partly due to the dependence of radiative sensitivity on
background climate states.
For the surface kernels, more prominent inter-kernel biases are found. For example, the
biases in the water vapor shortwave kernel in the Antarctic (Figure 7) and in the surface albedo
kernel in the Arctic (Figure 8) can reach 30%. It is especially noticed that some kernels have
considerably biased air temperature sensitivity values in the bottom atmospheric layers, which is
likely due to improper treatment in the perturbation experiments used for kernel computation.
The biases in both TOA and surface kernels discovered here affirm the importance of validating
the radiative sensitivity as noted by Huang and Wang (2019) and Pincus et al. (2020).
Applying the different kernels to quantifying the TOA and surface radiative feedbacks,
we compare the feedback differences caused by using different kernels and also the inter-GCM
spread of the feedback values (when measured by the same kernel). We find the kernel bias is
not a major cause of the inter-GCM feedback spread (Tables 3, 5). This finding is in consistency
with the previous assessments (e.g., Soden et al., 2008; Jonko et al., 2012; Vial et al. 2013).
Radiation closure tests show that the unexplained residuals are generally within 10% for
both TOA and surface analyses in terms of the global mean feedback, affirming the validity of
the kernels for feedback quantification for both budgets. This suggests that the large non-closure
residuals reported in some previous studies (e.g., Vargas Zeppetello, et al., 2019) are likely due
to kernel inaccuracy rather than the limitation of the kernel method. However, there are more
significant local non-closures, for example, in the shortwave in the Arctic region and around the
Antarctic continent, which is contributed, but cannot be fully explained, by the kernel
uncertainty. This points to the accuracy limit of the kernel (linear) method and calls for more
advanced methods, such as the neural network method (Zhu et al., 2019), for local feedback
analysis.

## Acknowledgements


We acknowledge the grants from the Natural Sciences and Engineering Research Council
of Canada (RGPIN-2019-04511) and the Fonds de Recherché Nature et Technologies of Quebec
(2021-PR-283823) that supported this research. H. Huang thanks Yonggang Liu, Jun Yang and
Qiang Wei for hosting her visit at Peking University and assisting her with computing resources.

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

**Appendix**

The ERA5 kernels are computed following Eq. (1) and the approach outlined in Section 2.2.

1. Surface variable kernels

To execute the partial radiative perturbation computations, the perturbations are prescribed as the following: for the 2D feedback variables, the surface temperature is increased by 1 K and the albedo is increased by 1% at each location. Hence, the units of the two kernels, $K_{Ts}$ and $K_{ALB}$ are W/(m$^2$ K)  and W/(m$^2$ 1%), respectively. When applying them to feedback quantification, their feedbacks are quantified as

$$\Delta R_{Ts} = K_{Ts} \cdot \Delta T_S \qquad (A1)$$
$$\Delta R_{Alb} = K_{ALB} \cdot \Delta Alb \qquad (A2)$$

where $\Delta T_S$ should be measured in the units of K and $\Delta Alb$ in percent, i.e., the multiply of 1%.

2. Water vapor kernel

For the 3D feedback variables, the perturbations are applied to each of the 37 pressure layers (from 1hPa to 1000hPa), one layer at a time and then normalized by the layer thickness. For the water vapor kernel, a 10% incremental perturbation of the water vapor concentration is used. To adapt to the convention used in the majority of the existing kernels, we convert the units of the kernels to represent the radiative flux change corresponding to an increase of water vapor concentration that conserves the relative humidity of the layer under a 1-K increase in air temperature, i.e., converting the units from W/(m$^2$ $\Delta q_0^{+10\%}$ 100hPa)  to W/(m$^2$ $\Delta q_0^{+1K}$ 100hPa):

$$K_q^{+10\%} = \frac{\Delta R_0}{\Delta q_0^{+10\%}} \qquad (A3)$$

$$K_q^{+1K} = \frac{\Delta R_0}{\Delta q_0^{+1K}} = K_q^{+10\%} \cdot \frac{\Delta q_0^{+10\%}}{\Delta q_0^{+1K}} = K_q^{+10\%} \cdot \frac{\Delta q_0^{+10\%}}{q_0} \cdot \frac{e_s(T_0)}{e_s(T_0+1K)-e_s(T_0)} \qquad (A4)$$

Where $q_0$ is the unperturbed water vapor concentration, in units of kg kg$^{-1}$. $\Delta q_0^{+10\%}$ is a 10% increment in water vapor concentration. $e_s(T)$ is the saturated water vapor pressure under temperature $T$, and can be measured by empirical formulas; hence, $\Delta q_0^{+1K}$ can be measured as $q_0[\frac{e_s(T_0+1K)}{e_s(T_0)} - 1]$. Accordingly, when the water vapor kernel is used for water vapor feedback quantification, the feedback is measured as:

$$\Delta R_q = K_q^{+1K} \cdot \Delta q^{+1K} = K_q^{+1K} \cdot \frac{\Delta q}{\Delta q_0^{+1K}} = K_q^{+1K} \cdot \frac{\Delta q}{q_0} \cdot \frac{e_s(T_0)}{e_s(T_0+1K)-e_s(T_0)} \qquad (A5)$$

where $\Delta q = q - q_0$ measures the change in water vapor concentration and is normalized by $\Delta q_0^{+1K}$ to give the factor that is multipliable with the $K_q^{+1K}$ kernel value. If using the Clapeyron-Clausius relation, the above expression can be further approximated as

$$\Delta R_q = K_q^{+1K} \cdot \frac{\Delta q}{q_0} \cdot \frac{e_s}{(de_s/dT) \cdot 1K} = K_q^{+1K} \cdot \frac{\Delta q}{q_0} \cdot \frac{R_v}{L_v} \cdot \frac{T_0^2}{1K} \qquad (A6)$$

where $R_v$ and $L_v$ are the gas constant and specific latent heat of water vapor, respectively. Note that when the kernels are used, $T_0$ and $q_0$ typically take their values from the base climate appropriate to the application, e.g., the unperturbed climate of a GCM experiment, not necessarily the dataset used for kernel computation.

Earth System Discussions
Open Access Science
Data

3. Air temperature kernel
For the air temperature kernel, to be consistent with the "inhomogeneous path treatment"
that accounts for the vertically non-uniform temperature distribution within each discrete
atmospheric layer (Mlawer et al., 1997), perturbations are added not only to the layer-mean
temperature but also the temperature at the exiting boundary of radiative fluxes of interest (i.e.,
the upper boundary of each layer for the TOA flux and the lower boundary for the surface flux),
to appropriately represent the physical temperature perturbation in each layer.
A meaningful test to affirm the validity of the air temperature kernel is a vertical sum test,
i.e., a linear additivity test to verify the vertical integration of the kernel values reproduce the
flux change, either at TOA or surface, in response to a whole-column air temperature increase of
1K. Figure A1 shows that the ERA5 kernel well passes this test. However, as shown by Figure
12, some kernels (e.g., HadGEM3 kernel) show much weaker radiative response at surface,
possibly due to improper treatment of the air temperature perturbation in the kernel computation,
which may lead to an underestimated air temperature feedback and large biases in the surface
feedback analysis.

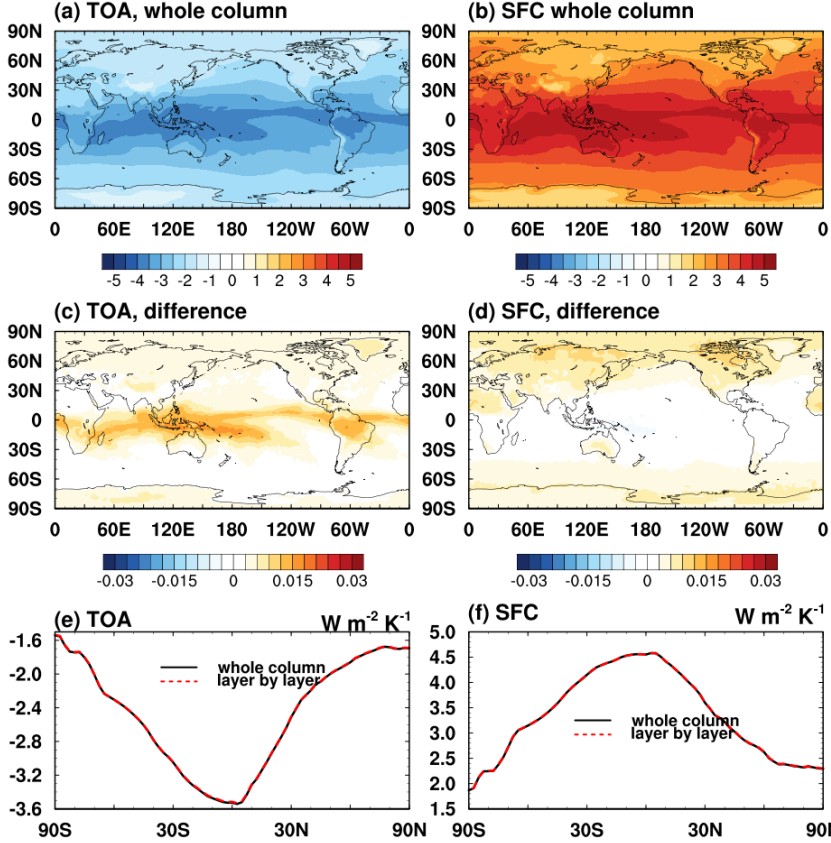

Figure A1. Monthly mean TOA and surface radiation flux change in response to a +1K
air temperature perturbation throughout the vertical column: (a, b) computed by a radiation



model, RRTMG; (c, d) difference of vertical sum of air temperature kernels compared to truth in
(a, b); (e, f) comparison of the zonal mean.
Another trickiness in the computation of air temperature kernel for surface flux is that the
surface in radiative transfer models is also the lower boundary of the lowermost atmospheric
layer. If the effects of the surface temperature perturbation on the emission of the surface and
that of the lowermost atmospheric layer are not distinguished, this may lead to improper
interpretation and use of the surface temperature kernel. In our ERA5 kernel, the two effects are
considered separately: according to radiative transfer theory, an increase in surface skin
temperature only affects the surface upward emission; an increase in air temperature only affects
the downward radiation. In some other kernels such as CAM5, these effects are not
distinguished, so that the kernel value represents the net effect, i.e., change in the sum of both
downward and upward. As a result, in Table 6, we can only report the sum of surface and air
temperature feedbacks. Figure A2 shows the comparison of vertically integrated air temperature
kernels and the sum of surface and air temperature kernels between ERA5, CAM5 and
HadGEM3. Although the strength of vertically integrated air temperature kernel for CAM5 is
much weaker than that for ERA5 (Figure A2a and b), the sum of surface and air temperature
kernel between these two datasets are in good agreement (Figure A2c and d). Another noticeable
feature in Figure A2 is that the HadGEM3 kernel shows an underestimation in vertical
integration of air temperature kernel and an overestimation in the sum of surface and air
temperature kernel, likely due to mistreatment of the bottom layer and accounting for the biased
surface feedback analysis as shown in Figure 12.

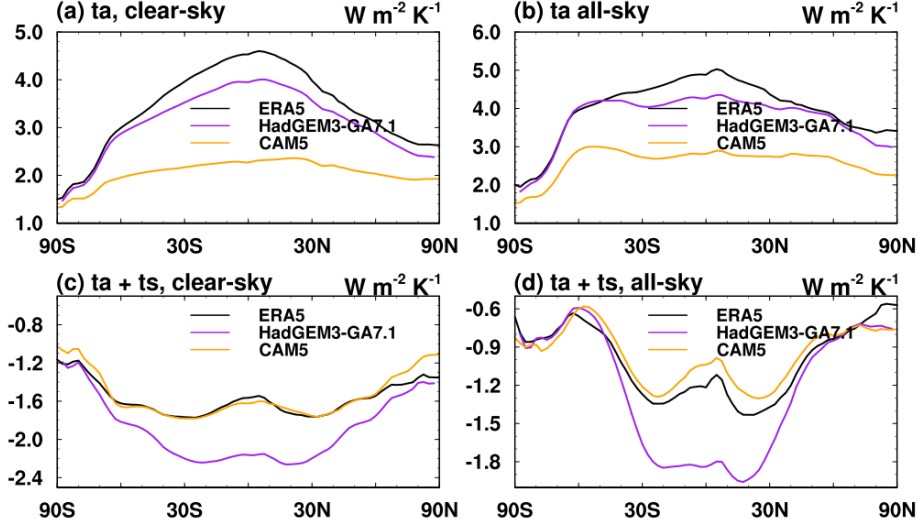

Figure A2. Comparison of annual mean kernels for ERA5, CAM5 and HadGEM3 for (a,
b) the vertically integrated air temperature kernel values, and (c, d) sum of surface and air
temperature kernels.
4. Time averaging



As described in Section 2.2, all the kernels provided for feedback analysis are averaged
from instantaneous kernel values over each calendar month and, in the ERA5 kernel, over
multiple years. This is to ensure proper sampling of radiative sensitivity values under different
atmospheric states, so that the kernels are representative of mean radiative sensitivity and thus
can be readily multiplied with monthly mean climate responses ($\Delta X$) to evaluate climate
feedbacks.
If the kernels are computed for fixed pressure levels, and if the pressure of any of these
levels of an instantaneous atmospheric profile is higher than the surface pressure (i.e., the level is
below the surface) at a time instance, this potentially creates inconsistency in the averaging
procedure. To address this concern, we set the kernel value to zero (as opposed to missing value)
before averaging. This is to ensure that when multiplied with the monthly mean climate response
($\Delta X$), the contribution of a pressure layer (e.g., that centered at 1000 hPa) is effectively counted
only for the fraction of time the layer exists (when surface pressure is higher than 1000 hPa).
Otherwise, the feedback quantification needs to be further weighted with fraction of time ($f$)
when the pressure layer exists. For example, if the surface pressure is larger than 1000hPa only
for half of time in a month ($f=0.5$), the radiation flux anomaly contributed by the layer centered
at 1000 hPa is:
$$\Delta R_{T_{1000hPa}} = K^*_{T_{1000hPa}} \cdot \Delta T_{1000hPa} \cdot f \qquad \text{(A7)}$$
Here, $K^*_{T_{1000hPa}}$ represents the kernel value averaged from the time instances when the layer
exists. Our averaging scheme is essentially to provide a kernel $K_{T_{1000hPa}} = K^*_{T_{1000hPa}} \cdot f$, so that
it can be simply multiplied with $\Delta T_{1000hPa}$ to obtain the same result.
Figure A3 illustrates the differences between $K^*_{T_a}$ and $K_{T_a}$, in terms of their vertically
integrated value. Such difference is pronounced over the Southern Oceans (around 60S), where
the surface pressure value varies considerably. This likely explains why Figure 3c and i show
noticeable differences in the air temperature kernel in this region.

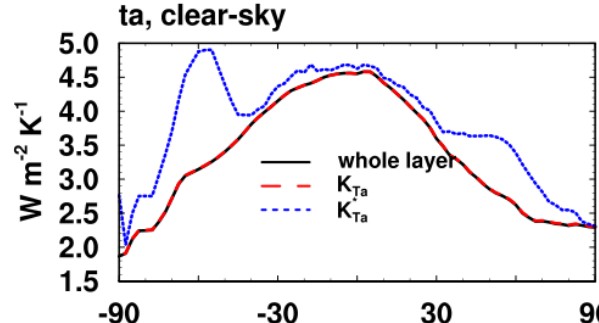




873    Figure A3. Zonal mean monthly mean air temperature kernels for surface flux from
874 ERA5 in clear-sky. Black line is the result from the whole column perturbation computation by
875 RRTMG, providing a "truth" for comparison. Red dashed line is the kernel weighted with
876 fraction of time ($K_{T_a}$) and blue dotted line represents results without weights ($K_{T_a}^*$).
878 5. Comparison of radiative kernels including all datasets
880    Figure A4 shows the comparison of air temperature kernel in all-sky including the Oslo
881 kernel, which compared with Figure 2k and l, shows greater discrepancies above the tropopause
882 and in lower troposphere.

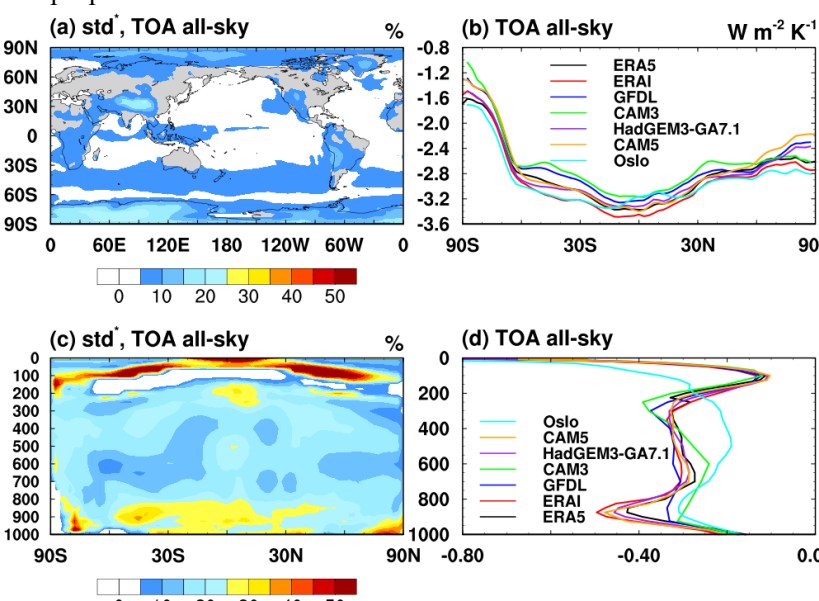

884    Figure A4. Similar to Figure 2, but including the Oslo kernel in all-sky. (a, c) fractional
885 discrepancies of the radiative kernels; (b) zonal mean vertically integrated radiative kernels,
886 units: W m$^{-2}$ K$^{-1}$; (d) global mean vertically resolved kernels, units: W m$^{-2}$ K$^{-1}$ 100hPa$^{-1}$.