# Peer review of "Radiative sensitivity quantified by a new set of radiation flux kernels based on the ERA5 reanalysis"

_Earth System Science Data, 2022_

## Referee Comment (RC1)

**Review of "Radiative sensitivity quantified by a new set of radiation flux kernels based on the ERA5 reanalysis" By Huang and Huang essd-2022-474**

**Summary**

Radiative kernels, which quantify the impact of unit changes in individual fields on radiative fluxes, have become a key tool in diagnosing radiative feedbacks both in climate models and in observations. In this study the authors develop a new set of radiative kernels using atmospheric and surface fields from the ERA5 reanalysis as inputs to the RRTMG radiation code. Unlike many previous kernels, they generate kernels for both the top-of-atmosphere (TOA) and the surface (SFC) such that impacts of changes in temperature, humidity, clouds, and surface albedo on surface radiation can be diagnosed. The ERA5 kernels are compared with previously-generated kernels, and inter-kernel differences are illuminated. The authors also explore the degree to which the derived kernels depend on the state of the climate, with input data from years impacted by El Nino events or with anomalous sea ice concentration resulting in kernels of different strength.

Overall I find the analysis to be solid and the presentation to be mostly clear. I have some suggestions for improving the readability of the paper and for presenting the relative importance of inter-kernel versus inter-model feedback differences. I also would like the authors to provide more evidence of the quality of this new kernel versus existing kernels. I recommend acceptance pending minor revision, as detailed below.

Mark Zelinka

**Major Comments**

- Abstract: Since the goals of this data journal are to publish work that documents useful datasets, with the scientific results being secondary, I felt that the abstract spent too much time on the inter-kernel comparison and not enough on the evaluation of the specific ERA5 kernels developed here. For example, it would be good to know in the abstract whether the new kernels have smaller residuals in the global mean or regionally than previous kernels. The bulk of the abstract describes results from all kernels collectively rather than focusing on the ERA5 kernels.
- The paper discusses TOA and SFC kernels but does not discuss the implied atmospheric kernels, derived via differencing the TOA and SFC kernels. Perhaps this would make the paper too long, but the authors might consider adding something on ATM kernels.
- Organization of the figures: I found it to be really taxing and distracting to have to jump between eight large figures on separate pages during Sections 3.1 to 3.2.
  - Section 3.1 discusses the ERA5 kernels in isolation. I think it would be more logical to have the first figure or two just show all the ERA5 kernels. This would include the first column of Figs 1-8, which is 32 panels. Perhaps you could have 2 figures with 4 rows and 4 columns each. This way a reader can see all of the new kernels just by looking at 2 figures, and can more easily match the discussion in Section 3.1 with the individual figure panels being discussed without flipping between 8 pages. If you do

this, I suggest re-labeling so it is obvious above each panel what one is looking at (i.e., "All-sky SFC Air Temperature Kernel", "Clear-sky TOA LW Water Vapor Kernel", etc.)

- Section 3.2 discusses the inter-kernel comparison, which refers solely to the two right columns of Figs 1-8. I would suggest making these their own figures. Perhaps some of this material could go in supporting information or the appendix, if you don't spend much time discussing it. Given the choice of journal, the focus of this manuscript should be to present and evaluate the new dataset, so this intercomparison is somewhat superfluous as it currently stands. It might be worth doing a more rigorous evaluation of ERA5 against other datasets rather than this discussion of the kernel differences collectively.
- Multi-kernel dataset: Have you considered doing the community service of placing the common-gridded multi-kernel dataset discussed on lines 294-296 on a public website?
- Throughout: The inter-kernel differences are referred to as "biases". Perhaps the authors are referring to the fact that all model-based kernels have a biased mean-state with respect to observations, but I think this verbiage is misleading. Also, the definition in L306 quantifies the bias with respect to the multi-kernel average, implying that the multi-kernel average is truth. The inter-kernel differences are a mix of model differences (in mean-state, radiation codes, etc.) and possibly the influence of actual biases (like the issues identified here in the HadGEM and Oslo kernels). If a kernel were to be built from a preindustrial control state, it may be less biased for computing feedbacks with respect to that state than the ERA5 kernels developed here; it depends on the context whether a given kernel is biased. I suggest changing all instances of "bias" to "differences" unless it can be shown to be a true bias with respect to a correct value.
- Tables 3-6: Could these results be presented more effectively? I'm not sure how insightful it
  is to present all the individual model results in four big tables. The message you are trying
  to convey is the relative importance of inter-kernel differences versus inter-model
  differences in SFC and TOA feedbacks, either broken down into LW, SW, and net, or into
  individual feedback components. I wonder if something analogous to Figure 1 of Chao and
  Dessler (2021) might be more effective. In this case, you would show the spread in each
  feedback from inter-kernel vs inter-model differences. Or would simply showing a figure
  comparing inter-kernel and inter-model standard deviations (ignoring the multi-model
  mean values) be more effective? Deciding on the most important points and then creating a
  figure that supports those points clearly would be worthwhile. Right now it is a bit hard for
  the reader to wade through these four big tables and extract the messages.
- In the end it is still a little unclear to me whether the new ERA5 kernel has a smaller residual than the other kernels. Can you make a stronger case for why we need this new kernel, and whether it is more accurate? Figures 11 and 12 suggest to me that the residuals are comparable to previous kernels; but this should be noted explicitly. If it is not more accurate, why should I use it over previous kernels? If it is more accurate, do you advocate that the community use this instead of the others? I think it is well established here and elsewhere that the inter-kernel differences are small relative to inter-model spread; why are we regularly making new kernels in this case?

**Minor Comments**

- Verbiage: Throughout the paper, I found some of the verbiage to be unnecessarily longwinded. Could "kernel of the surface flux" be the "surface kernel", for example?
- L22: "in" should be "for"
- L32: I don't understand what is meant by "inter-kernel bias-induced uncertainty", which appears in slightly modified phrasing in other places as well (L557). Is this just "inter-kernel differences"?
- L38: delete "on the other hand"
- L60: suggest also citing the recent work of Chao and Dessler (2021)
- L75: Suggest citing some additional work, some of which includes surface and atmosphere cloud radiative kernels (Zhang et al., 2021; Zhou et al., 2022, 2013)
- L81: suggest specifying "largely insensitive"
- L83: "are" should be "is"
- L107: suggest simplifying to "we intercompare"
- L109-111: suggest rephrasing this sentence, which I found hard to parse. Also, you probably want to specify that you are comparing across-model vs across-kernel differences in this sentence (I think)
- L150-152: I'm confused by how you describe the analysis. I thought kernels were constructed using one experiment, performing many calls to the radiative transfer code, each time with a single field / level / location perturbed. This is not how the procedure is described here.
- L168-169: Probably want to remind the reader why the factors of 4 and 8 are present in these expressions. It is because the radiation calculations are done 4- or 8-times daily, I think.
- L168: "kernels" should be singular
- L190: suggest "upwelling" instead of "outgoing". Also, suggest simplifying to "the kernel is negative"
- L206: should be "(f,I)" rather than "(g,I)"
- L253: "reduce" should be plural
- L257: I think you should specify that you are talking about the clear-sky TOA kernel here.
- L336: "by the inconsistency in" should be "by inconsistencies in"
- L343-344: could this be simplified to "state-dependency in the kernels"?
- L354-355: "the" before "interannual" and "cloudiness" is not needed
- L359: what is meant be "seasonal SST anomalies" Previously, it is stated that you are examining annual means.
- L363: "since" should be "in the"
- L364: "exemplify" should be "illustrate" or "highlight"
- L365: All sky what? Kernels?
- L370: I think some explanation of this result is warranted. Why does Figure 9e have that structure, wherein some regions that are moister and cloudier have a larger SW WV kernel but some do not (NE Pacific). Also, the panel titles in Figure 9 are a little ambiguous; suggest explicitly stating what is shown in each.

- Figure 10: suggest deleting the longitude labels which clutter the figure and seem unnecessary given the provided coastlines.
- L385-394: More explanation of why you get these results is needed. Also, this is too long of a sentence.
- L390-394: Is one of the take-aways here that it may be necessary to average over multiple years when constructing kernels? Or at least that one has to be careful not to choose a year with an extreme Nino index or huge sea ice anomalies when constructing kernels? You might consider making this point explicitly.
- L403: missing space between "Table" and "2"
- Table 2: "model top level" is not an accurate description of what is reported in that column
- L412-419: I think more description and motivation for using these experiments is needed. The abrupt-4xCO2 experiment is a fully-coupled experiment whereas piClim-4xCO2 is an atmosphere-only experiment. You should also cite the relevant piClim-4xCO2 experiment description paper (Pincus et al., 2016). I've never seen these two experiments differenced in order to derive the temperature-mediated responses without the confounding effects of rapid adjustments; this is clever although it limits the number of models available to analyze. (Although more than just 6 models are available as far as I can tell.) I suggest explaining these choices a little better. I would also suggest mentioning this methodological difference when coming your values to those of Zelinka et al (2020) – that study used piControl simulations as the baseline and computed abrupt-4xCO2 anomalies and feedbacks differently. It is reassuring that the results of the two approaches agree as well as they do.
- L445-446: The end of this sentence is redundant with previous statements; suggest deleting.
- L460, L467: small relative to what?
- L477: Suggest stating the name of the row rather than making the reader count.
- L478-480: suggest citing some examples to explain how you arrive at these percentage numbers. Are you comparing inter-kernel standard deviations to inter-model standard deviations?
- L488: these numbers seem misleading, because most feedbacks have roughly the same absolute value of inter-kernel spread; they just vary in the central value. If all feedbacks had the same inter-kernel spread, but one feedback happened to be zero (e.g., if the SW cloud amount feedback perfectly compensated a SW cloud albedo feedback), the inter-kernel spread relative to this would be infinite, but that is not really meaningful.
- L541: Delete "First of all"
- L583: This sentence seems to run on and should probably be broken up for clarity.
- L585-591: this sentence is also way too long and should be broken up
- L594: "it is especially noticed that" can be deleted
- L599-600: suggest making this more concise by removing redundancy
- L601-602: How could inter-model spread come from inter-kernel spread? Please rephrase.
- L603: rephrase to "finding is consistent with previous"
- L762: specify whether this is an absolute or relative change. I'm pretty sure it is the former.
- L767: not sure what is meant by the last phrase
- L818: "trickiness" is probably too informal; suggest "challenge"

- L835: I don't understand what is meant from "and accounting" onward
- Figure A2: are these SFC or TOA kernels? I assume SFC.
- L854: specify "in these cases"

**References**

- Chao, L.-W., Dessler, A.E., 2021. An Assessment of Climate Feedbacks in Observations and Climate Models Using Different Energy Balance Frameworks. J. Clim. 34, 9763–9773. https://doi.org/10.1175/JCLI-D-21-0226.1
- Pincus, R., Forster, P.M., Stevens, B., 2016. The Radiative Forcing Model Intercomparison Project (RFMIP): experimental protocol for CMIP6. Geosci. Model Dev. 9, 3447–3460. https://doi.org/10.5194/gmd-9-3447-2016
- Zelinka, M.D., Myers, T.A., McCoy, D.T., Po-Chedley, S., Caldwell, P.M., Ceppi, P., Klein, S.A., Taylor, K.E., 2020. Causes of Higher Climate Sensitivity in CMIP6 Models. Geophys. Res. Lett. 47, e2019GL085782. https://doi.org/10.1029/2019GL085782
- Zhang, Y., Jin, Z., Sikand, M., 2021. The Top-of-Atmosphere, Surface and Atmospheric Cloud Radiative Kernels Based on ISCCP-H Datasets: Method and Evaluation. J. Geophys. Res. Atmospheres 126, e2021JD035053. https://doi.org/10.1029/2021JD035053
- Zhou, C., Liu, Y., Wang, Q., 2022. Calculating the Climatology and Anomalies of Surface Cloud Radiative Effect Using Cloud Property Histograms and Cloud Radiative Kernels. Adv. Atmospheric Sci. 39, 2124–2136. https://doi.org/10.1007/s00376-021-1166-z
- Zhou, C., Zelinka, M.D., Dessler, A.E., Yang, P., 2013. An Analysis of the Short-Term Cloud Feedback Using MODIS Data. J. Clim. 26, 4803–4815. https://doi.org/10.1175/JCLI-D-12-00547.1

---

## Referee Comment (RC2)

The paper presents a set of newly calculated radiation flux kernels using the ERA5 reanalysis dataset. The authors discuss how the new radiation flux kernels differ from previous ones and how they can be used to improve our understanding of Earth's climate system. Overall, this paper presents a valuable contribution to the field of climate science by providing a new set of radiation flux kernels that can help improve our understanding of Earth's climate sensitivity. I have several major concerns and recommend a major revision.

1. In recent years, one of the improvements of radiative kernels is the development of radiative kernels at the surface (SFC) and in the atmospheric column. The kernels at SFC have been calculated not only from reanalysis data but also from observational data (Karmer et al. 2019). Although the ERA5-derived kernels show high consistency with model-based kernels, feedback parameters obtained from model- and reanalysis-based kernels have large discrepancies with observation-based feedback parameters, especially for the cloud feedback (Karmer et al. 2019; Zhang et al. 2021). Would you like to conduct more analysis and add more discussion on the differences in cloud feedbacks derived from various data sources?

2. Cloud feedbacks are diagnosed using the adjusted cloud radiative effect method by assuming that all-sky decomposition has the same non-closure residual. There are some flaws in the assumption. First, the residual ($res^o$) is introduced during the single variable perturbation or linear decomposition without involving cloud related process. Second, the all-sky decomposition is assumed that has the same non-closure residual with clear-sky ($res^o = res^c$). It should be proved before being applied. Once the cloud related processes are introduced, it would be nearly impossible for the non-closure residual in all-sky to be same as the residual in clear-sky. Please reconsider Eqs. 5-6.

3. The non-closure residual terms due to nonlinear effect are discussed in Figs. 11 and 12. As shown in Fig. 11, the residual term at the TOA mainly arises from shortwave radiation over regions with abundant sea ice cover. Huang et al. (2021) pointed out that the nonlinear effects are resulted from the coupling effect between the surface

albedo and cloud, and between the air temperature and cloud. Given the significant interactive between cloud and other climate variables, it's inappropriate to assume the same residual between all-sky and clear-sky conditions. For the residual term at the SFC (Fig. 12), the magnitude of longwave radiation is comparable to the magnitude of shortwave. There is a lack of necessary discussion of the increase in LW residual at SFC relative to that at TOA.

4. The most important issue is that what's the contribution of ERA5-based kernel to the radiative kernel method. It's highly consistent with model simulation-based kernel, while model simulation can be applied to more accurate analysis such as diagnostic analysis on the role of dynamic processes in climate response.

5. The order of the figures needs to be adjusted. It would be better to cite figures near the context instead of figures far away from the context.

6. In Fig. 6b, the fractional discrepancies of the sensitivity of the TOA SW flux to water vapor in the tropics show six large value centers from the east coast of Africa to the equatorial eastern Pacific. It's hard to understand these large value centers physically. Could you explain it?

**References**

Kramer, R. J., A. V. Matus, B. J. Soden, and T. S. L'Ecuyer, 2019: Observation-based radiative kernels from CloudSat/CALIPSO. *J. Geophys. Res. Atmos.,* 124, 5431–5444, https://doi.org/10.1029/2018JD029021.

Zhang, Y., Z. Jin, and M. Sikand, 2021: The top-of-atmosphere, surface and atmospheric cloud radiative kernels based on ISCCP-H Datasets: Method and evaluation. *J. Geophys. Res. Atmos.,* 126, 1–34, https://doi.org/10.1029/2021JD 035053.

---

## Referee Comment (RC3)

In this manuscript, the authors introduce a new set of all-sky and clear-sky, top-of-atmosphere (TOA) and surface radiative kernels, generated with the RRTMG radiative transfer model for 5 years of input fields from ERA5 reanalysis data. The authors incorporate these kernels into a more general inter-comparison of the magnitude and structure of existing sets of radiative kernels, highlighting the sensitivity of radiative kernels to the input climate state fiends used to compute them. Along these lines, they also highlight the sensitivity of radiative kernels to inter-annual variability in the climate state, taking advantage of the fact that the ERA5 kernels have been computed for 5 years of data, over a period with notable changes in ENSO and sea ice coverage.

The manuscript is comprehensive and well written. More multi-kernel comparison analysis is certainly needed, so this will be a welcomed addition to the literature. However, I feel the paper suffers in a few ways by trying to balance all three tasks (introduce a new kernel, multi-kernel comparison, sensitivity of kernels to variability) in one paper. For instance, I think the analysis of inter-annual variability is the most valuable part of the work, but the analysis is not well connected to the multi-kernel comparison, and the analysis is not as in depth as it could be. Additionally, it's not clear why we need the ERA5 kernels when they don't seem all that different from the others, and the kernels based on ERA-Interim reanalysis data (ERAi) previously developed by the second author of this paper were also calculated for 5 years of data and could have been used for the inter-annual variability analysis instead. Given the journal, I think justifying the new data product is important. My comments below reflect these concerns and hopefully add some additional, useful explanation to my points. Given these, I think the paper deserves consideration for publication, pending major revisions.

Ryan Kramer

**General**

1) As noted above, some additional justification for producing the ERA5 kernels is necessary, particularly given that the ERAi kernels exist, which use similar input data and a similar radiative transfer code, also for 5 years of data. For example, is the improvement of RRTMG (ERA5 kernels) over RRTM (the ERAi kernels) large enough to warrant new kernels? Even if so, the sensitivity of kernels to RT is not really a focus of the analysis here. Does ERA5 have more realistic climate state fields than ERAi? The two kernels were computed over different periods, so maybe that is reason to make new kernels? But the period for the ERAi kernels (2008-2012) also has notable swings in ENSO, so I don't quite see the advantage of the later period used for ERA5 kernels.

    1a) Related, what is the justification for developing radiative kernels from reanalysis when the fields are available from models and observations? Arguably reanalysis offers a happy middle between the two. They may not be pure observations, but they do have the full diurnal cycle that most satellite observations do not have. This may be important for diagnosing feedbacks in models, where the model fluxes are also a response to the full diurnal cycle. But

there is also the argument that, in order to diagnose the true feedback, model feedbacks should be diagnosed with a kernel developed from models and observed feedbacks should be diagnosed with appropriate observations. What is the value of reanalysis-based kernels in that context? Some discussion along these lines would be really valuable to a community often confused about what kernels they should be using.

2) This may be the only example where an "older" (ERAi) and "updated" (ERA5) radiative kernel were developed by the same research group using similar RT codes. This could be a really powerful tool for the multi-kernel comparison analysis and should be exploited here, but has not been yet. If the second author still has access to the ERAi kernel input data, I would like this team to include a more in-depth comparison of the ERAi and ERA5 kernels in the context of the multi-kernel intercomparison. Throughout the current manuscript, the authors highlight examples of large multi-kernel differences, tying them to potential differences in the underlying climate input data. For a given example, is the spread also evident in differences between the ERAi and ERA5 kernels? If so, the authors should analyze the climate input fields directly to reveal specifics about why the kernels differ. A few specific comments in the section below try to prompt this type of analysis. Among other groups, I think this could be extremely useful for the ECMWF developers of ERAi and ERA5, who are always trying to understand the biases and limitations of their product, giving your work exposure to an additional, large community.

3) Section 3.1: The authors should rethink the presentation and the focus of discussion in this section. First, general descriptions of the sign, basic explanation of the causes of the sign, and the zonal-mean vertical structure of kernels are discussed here for the new ERA5 kernels, but these topics have been covered extensively for other kernels and the new kernels don't seem to deviate from that. Therefore, it seems redundant to repeat that information here. This is true even for surface radiative kernels, where the structure and sign were covered by Kramer et al. 2019 a and b (see refs below). The description of the horizontal spatial structure of the kernels is newer however, and worthy of focus in this section. Second, the number of figures and figure panels in this section is also overwhelming for the reader and should be consolidated. Given these points, I would instead:

-For Figures 1-8, make the first figure or two just the spatial maps of each kernel, with a title for each subplot that describes which kernel we are looking at (e.g. all sky, surface temp, clear-sky SW WV, etc.).
-Given their prevalence elsewhere (e.g. Soden et al. 2008; Block and Mauritsen 2013; Kramer et al. 2019a,b, Smith et al. 2021), the latitude-pressure subplots of the ERA5 kernels can be combined and put in supplemental material.
-Any subplot currently referring to the intercomparison across existing kernels should be saved for new, separate figures placed in Section 3.2, where that material is discussed in the text.

This above list is just a recommendation. I'm sure there are other ways of reorganizing the plots to improve manuscript readability, but some reorganization is necessary.

*Kramer, R. J., A. V. Matus, B. J. Soden, and T. S. L'Ecuyer, 2019: Observation-Based Radiative Kernels From CloudSat/CALIPSO. JGR Atmospheres, 124, 5431–5444, https://doi.org/10.1029/2018jd029021.*

*Kramer, R. J., B. J. Soden, and A. G. Pendergrass, 2019: Evaluating Climate Model Simulations of the Radiative Forcing and Radiative Response at Earth's Surface. Journal of Climate, 32, 4089–4102, https://doi.org/10.1175/jcli-d-18-0137.1.*

4) It is evident that clouds play an important role in determining the spatial pattern of the all-sky radiative kernels. More description of the type of clouds impacting the kernels would be very useful and novel. For example, cloud vertical extent? Cloud base height (for sfc kernels)? Optical properties? A deeper analysis of the ERA5 cloud fields would be helpful here. And if the fields are still available for the ERAi kernels, even better.

5) I think the analysis of inter-annual variability in the kernel is the most interesting contribution of this paper to the literature. It deserves a more prominent place in the title, abstract, and conclusion section.

6) The inter-annual variability analysis feels disjointed from the introduction of the new kernels in Section 3.1, the multi-kernel comparison, and the feedback estimate section. It is evident that the ERA5 kernels are sensitive to inter-annual variability, but does this really matter for overall kernel spread? For instance, the ERA5 all-sky TOA Ts kernel is clearly sensitive to interannual variability in the Eq. Pacific at ~Longtiude 180, but this doesn't appear to be a particularly noteworthy area of inter-kernel differences in fig 1e. It may very well be important, but the analysis at present doesn't offer enough of a connection to make that point. Comparing the inter-annual variability in the ERA5 vs ERAi kernels in more detail may be a useful starting point to help make the connection between this section and the others.

7) After the interesting analysis showing the sensitivity of kernels to inter-annual variability spatially, the feedback analysis in Section 4.2 and 4.3 mostly just focuses on global-mean values. While this type of analysis is valuable in a general sense for kernel users, it doesn't quite fit well with this paper, particularly because the ERA5 kernels don't stand out as being more accurate or unique. Instead, I'd like to authors to focus more on multi-kernel differences in the feedback spatial patterns. Given the large focus on the pattern effect and the influence of regional feedbacks on the global-mean in recent years, I think that could be particularly citeable. We know kernels are generally in agreement in the global-mean (especially for the TOA). But what about for kernel spread in estimates of the regional feedbacks?

8) It is tough to pick out valuable information from your tables of TOA and Surface feedbacks. The authors should turn those into summary figures (e.g. dot plots or scatter plots used by e.g. Smith et al. 2018 supplemental, or Zelinka et al. 2020) where possible.

9) There are now many observation-based kernels in the literature from CloudSat/CALIPSO observations, CERES CCCM products, AIRS, and a bunch of others specific to surface albedo kernels that are not included in the multi-kernel analysis here.  I think it's an open question whether these observational kernels should be included in a comparison with model-based kernels or not.  The authors should provide a brief reason for not incorporating them (or should include them if they feel its appropriate), especially since the Kramer et al. and Thorsen et al. reference papers are cited in the text.

**Specific or Minor Comments**

Line 311-312 and Appendix: We discussed similar RT issues regarding the surface temperature kernel for surface fluxes in Kramer et al. 2019 (JClim).  The authors can cite and/or refer to it for some additional support.  We also argued that, as noted in the author's current Appendix and elsewhere, similar RT issues can bias the lowest level of the surface flux Ta kernel in an equal and opposite manner as the Ts kernel, thereby allowing a kernel like CAM5 to be correct in its estimate of the vertically integrated Temperature feedback, but for the wrong reason.  This likely explains why CAM5 and ERA5 kernels agree in Figure A2c and A2d. Your text somewhat gets to this point, but I'd call it out directly as a warning to kernel users: Some kernels may give you the correct temp. feedback for the wrong reason.  Presumably this is true for the TOA temperature feedback too, but the contribution from the surface and near surface layers to that vertically integrated feedback are small, so maybe it doesn't matter much?

*Kramer, R. J., B. J. Soden, and A. G. Pendergrass, 2019: Evaluating Climate Model Simulations of the Radiative Forcing and Radiative Response at Earth's Surface. Journal of Climate, 32, 4089–4102, https://doi.org/10.1175/jcli-d-18-0137.1.*

Line 316-319 and more generally: Bright and O'Holleran (2019) and Donohoe et al. (2020) performed nice, detailed comparisons of surface albedo kernels. These papers should be cited and their work should be put into context of the author's own results within the present manuscript. This would also be useful for the section on diagnosing radiative feedback spread, since the authors show that the kernels give quite different results in the poles. Riihela et al. (2021) also did a comparison of surface albedo kernels in the context of sea ice states, and should be cited somewhere in the present manuscript.

*Bright, R. M., and T. L. O'Halloran, 2019: Developing a monthly radiative kernel for surface albedo change from satellite climatologies of Earth's shortwave radiation budget: CACK v1.0. Geosci. Model Dev., 12, 3975–3990, https://doi.org/10.5194/gmd-12-3975-2019.*

*Donohoe, A., E. Blanchard-Wrigglesworth, A. Schweiger, and P. J. Rasch, 2020: The Effect of Atmospheric Transmissivity on Model and Observational Estimates of the Sea Ice Albedo Feedback. Journal of Climate, 33, 5743–5765, https://doi.org/10.1175/jcli-d-19-0674.1.*

*Riihelä, A., R. M. Bright, and K. Anttila, 2021: Recent strengthening of snow and ice albedo feedback driven by Antarctic sea-ice loss. Nat. Geosci., 14, 832–836, https://doi.org/10.1038/s41561-021-00841-x.*

Line 316-227 and more generally: Aligning with general comment 4 above, the author's assumption that clouds are the cause of the discrepancies discussed here is likely right, but this has been alluded to before in past work. The author's have a unique opportunity to prove it directly by using the kernel's cloud input data directly in the analysis. Using the ERA5 and ERAi cloud fields can the authors confirm their assumption that cloud fields mattter? Or provide a more detailed analysis? Among all the potential sources of kernel differences, clouds seem to warrant deeper investigation.

Figure 2f: The ERAi and ERA5 kernels are noticeably different below ~800mb. Why?

Figure 3e and 3k: There are large standard deviations relative to the magnitude of the ERA5 kernels at certain locations within the vertical kernel structure, but for anything above ~950mb, the kernel magnitude is small relative to the lowermost atmospheric levels, and likely does not contribute much to the vertically-integrated quantity. A zoomed in version of these plots, highlighting the important standard deviation across kernels in the lowermost levels, would be informative.

Figure 3f and 3l. Why is the ERAi kernel so much larger at the lowest levels near the surface than the ERA5 kernel (and the other kernels)? This is true for both all-sky and clear-sky. Could vertical resolution of the kernels be playing a role? We discuss the potential influence of resolution in the Appendix of Kramer et al. (2019, JClim).

Interestingly, there is also a fairly large difference between ERA5 and ERAi in the all-sky LW WV kernel for surface fluxes (figure 5L), but it only shows up in the all-sky kernel, not the clear-sky. Does that suggest clouds matter more for the LW WV kernel in explaining differences than they do for the LW Ta kernel, relative to other potential sources of kernel spread?

Line 368: Is this the sensitivity of the surface to the vertically integrated kernel change? Only the sensitivity to a certain level of WV change? It is not obvious from the figure 9 caption either. Some rethinking of how the kernels are described in the text here and elsewhere would be helpful. Maybe shorthand acronyms or some other naming convention would be helpful.

Plot 9: I really like this figure. I'm not sure anyone else has shown the sensitivity of these temperature and water vapor kernels to variability spatially yet. But I think it would be helpful to go one step further and show what level of WV and cloud variability is impacting the temporal variability of the kernels most. And does that particular level help explain the multi-kernel spread in those kernels, evident in Figure 1-8? Or is inter-annual variability not enough to explain the kernel spread? This comments connects with my general comment #6 above.

Plot 9: The TOA SW WV kernel is a potentially interesting case where the largest multi-kernel spread (tropics around 500mb in Figure 6k) sits above the level at which the ERA5 kernel is the strongest (closer to 800mb in Figure 6j).  Building on my comment above, how does the importance of inter-annual variability play into the large kernel spread at this ~500mb level?  And does the spread at 500mb actually matter much for the vertically-integrated quantity?  This level of detail could be useful for e.g. modeling centers trying to connect TOA biases to particular biases in their climate states.

Figure 10: I struggle to see spatially where the variability in water vapor is having an effect on the kernels shown in the other subplots.  Maybe only for the NE coast of Greenland?  Should cloud changes be shown in this plot instead?  I'd give more detailed evidence about why water vapor is important here.

Line 412-419: Some explanation of why you need both abrupt4xCO2 and piCLim-4xCO2 (e.g. to remove rapid adjustments) is needed, since most people just use abrupt4xCO2 with Gregory regression to get feedbacks.

Line 436 and Equation 5: What does the clear-sky residual term mean physically and why does it matter for computing cloud feedback?  Although the author's math in Equation 6 works out to be the same as what everyone else uses, I think the way they've introduced this method, and terminology used, is less common. Some extra detail would be helpful.

Line 463-464: Block and Mauritsen (2013) can be cited here for their nice discussion and analysis of the non-linearity of the surface albedo kernel in 4xCO2 runs.

*Block, K., and T. Mauritsen, 2013: Forcing and feedback in the MPI-ESM-LR coupled model under abruptly quadrupled CO2. J. Adv. Model. Earth Syst., 5, 676–691,*
*https://doi.org/10.1002/jame.20041.*

Last Column Table 3:  The multi-model values differ somewhat across the different kernels, but not the associated standard deviations, which look to be essentially the same for all rows of the column.  Does this suggest the kernels can estimate feedbacks differently in a systematic manner across all models, but they do not necessarily estimate the magnitude of the feedback spread differently?  In other words, the kernel may get the model-mean feedback value wrong, but the spread in feedbacks correct?  This is worth noting if so.

Line 487-491: Since you are not using cloud radiative kernels, the inter-kernel differences in cloud feedback must come from the difference between all-sky and clear-sky kernels (cloud masking). Can you identify which of the kernel terms is the culprit? From a related discussion see text around figs 9-11 in Kramer et al. (2019, JGR), for example.

Kramer, R. J., A. V. Matus, B. J. Soden, and T. S. L'Ecuyer, 2019: Observation-Based Radiative Kernels From CloudSat/CALIPSO. JGR Atmospheres, 124, 5431–5444,
https://doi.org/10.1029/2018jd029021.

Line 769-793:  This is a really nice description of how you develop the water vapor kernel. I'd highlight in the main text that you have included this section in appendix. I think many kernel users and developers are looking for a description like this and will turn to it in the future. There are often question about this calculation.

---

## Author Comment (AC1)

**Response to Reviewer Comments**

We thank the reviewer for thoughtful and helpful comments. Below are our responses (in regular font) to the comments (in ***bolded italic*** font).

*Reviewer #1:*
*Review of "Radiative sensitivity quantified by a new set of radiation flux kernels based on the*
*ERA5 reanalysis"*
*By Huang and Huang*
*essd-2022-474*
*Summary*

*Radiative kernels, which quantify the impact of unit changes in individual fields on radiative*
*fluxes, have become a key tool in diagnosing radiative feedbacks both in climate models and in*
*observations. In this study the authors develop a new set of radiative kernels using*
*atmospheric and surface fields from the ERA5 reanalysis as inputs to the RRTMG radiation*
*code. Unlike many previous kernels, they generate kernels for both the top-of-atmosphere*
*(TOA) and the surface (SFC) such that impacts of changes in temperature, humidity, clouds,*
*and surface albedo on surface radiation can be diagnosed. The ERA5 kernels are compared*
*with previously generated kernels, and inter-kernel differences are illuminated. The authors*
*also explore the degree to which the derived kernels depend on the state of the climate, with*
*input data from years impacted by El Nino events or with anomalous sea ice concentration*
*resulting in kernels of different strength.*

*Overall I find the analysis to be solid and the presentation to be mostly clear. I have some*
*suggestions for improving the readability of the paper and for presenting the relative*
*importance of inter-kernel versus inter-model feedback differences. I also would like the*
*authors to provide more evidence of the quality of this new kernel versus existing kernels. I*
*recommend acceptance pending minor revision, as detailed below.*

*Mark Zelinka*

*Major Comments*

*• Abstract: Since the goals of this data journal are to publish work that documents useful*
*datasets, with the scientific results being secondary, I felt that the abstract spent too much time*
*on the inter-kernel comparison and not enough on the evaluation of the specific ERA5 kernels*
*developed here. For example, it would be good to know in the abstract whether the new*
*kernels have smaller residuals in the global mean or regionally than previous kernels. The*
*bulk of the abstract describes results from all kernels collectively rather than focusing on the*
*ERA5 kernels.*

Revised. ERA5 TOA kernels are as good as other kernel datasets while for surface kernels,
ERA5 kernels show better performance, in terms of the radiative sensitivity and radiation closure
test. The revised abstract emphasized this point.

*• The paper discusses TOA and SFC kernels but does not discuss the implied atmospheric*
*kernels, derived via differencing the TOA and SFC kernels. Perhaps this would make the*
*paper too long, but the authors might consider adding something on ATM kernels.*

Agreed: the ATM kernels are as important as TOA and SFC kernels. Considering the length and readability of the manuscript, we added ATM kernel results in the supplement.

*• Organization of the figures: I found it to be really taxing and distracting to have to jump between eight large figures on separate pages during Sections 3.1 to 3.2.*

*Section 3.1 discusses the ERA5 kernels in isolation. I think it would be more logical to have the first figure or two just show all the ERA5 kernels. This would include the first column of Figs 1-8, which is 32 panels. Perhaps you could have 2 figures with 4 rows and 4 columns each. This way a reader can see all of the new kernels just by looking at 2 figures, and can more easily match the discussion in Section 3.1 with the individual figure panels being discussed without flipping between 8 pages. If you do this, I suggest re-labeling so it is obvious above each panel what one is looking at (i.e., "All-sky SFC Air Temperature Kernel", "Clear-sky TOA LW Water Vapor Kernel", etc.)*

*Section 3.2 discusses the inter-kernel comparison, which refers solely to the two right columns of Figs 1-8. I would suggest making these their own figures. Perhaps some of this material could go in supporting information or the appendix, if you don't spend much time discussing it. Given the choice of journal, the focus of this manuscript should be to present and evaluate the new dataset, so this intercomparison is somewhat superfluous as it currently stands. It might be worth doing a more rigorous evaluation of ERA5 against other datasets rather than this discussion of the kernel differences collectively.*

We reorganized the figures, with the ERA5 kernel now shown in Figure 1-2 and the comparison with other datasets in Figure 3-4. We keep the comparison of all kernel datasets in Section 3.2 (e.g., the fractional discrepancies) as oppose the difference of ERA5 kernels against other datasets, as there is no truth value to be compared with and the point in this section is to show where these datasets differ most, and indeed the comparison reveals some issues in current SFC kernels.

*• Multi-kernel dataset: Have you considered doing the community service of placing the common-gridded multi-kernel dataset discussed on lines 294-296 on a public website?*

We added it in the data repository.

*• Throughout: The inter-kernel differences are referred to as "biases". Perhaps the authors are referring to the fact that all model-based kernels have a biased mean-state with respect to observations, but I think this verbiage is misleading. Also, the definition in L306 quantifies the bias with respect to the multi-kernel average, implying that the multi-kernel average is truth. The inter-kernel differences are a mix of model differences (in mean-state, radiation codes, etc.) and possibly the influence of actual biases (like the issues identified here in the HadGEM and Oslo kernels). If a kernel were to be built from a preindustrial control state, it may be less biased for computing feedbacks with respect to that state than the ERA5 kernels developed here; it depends on the context whether a given kernel is biased. I suggest changing all instances of "bias" to "differences" unless it can be shown to be a true bias with respect to a correct value.*

Revised. In equation (2), we use the multi-kernel mean as a reference value to illustrate how the
kernel values vary among dataset, rather than deeming it as a "truth" value. We add a note in
Line 279-280 to explain it.
*• Tables 3-6: Could these results be presented more effectively? I'm not sure how insightful it*
*is to present all the individual model results in four big tables. The message you are trying to*
*convey is the relative importance of inter-kernel differences versus inter-model differences in*
*SFC and TOA feedbacks, either broken down into LW, SW, and net, or into individual*
*feedback components. I wonder if something analogous to Figure 1 of Chao and Dessler*
*(2021) might be more effective. In this case, you would show the spread in each feedback from*
*inter-kernel vs inter-model differences. Or would simply showing a figure comparing inter-*
*kernel and inter-model standard deviations (ignoring the multi-model mean values) be more*
*effective? Deciding on the most important points and then creating a figure that supports*
*those points clearly would be worthwhile. Right now it is a bit hard for the reader to wade*
*through these four big tables and extract the messages.*
We reorganized these results and put the tables of component feedback parameters to the
supplement for readers who are interested and used figure 8 and 10 to show the relatively larger
inter-model difference than inter-kernel difference.
*• In the end it is still a little unclear to me whether the new ERA5 kernel has a smaller*
*residual than the other kernels. Can you make a stronger case for why we need this new*
*kernel, and whether it is more accurate? Figures 11 and 12 suggest to me that the residuals*
*are comparable to previous kernels; but this should be noted explicitly. If it is not more*
*accurate, why should I use it over previous kernels? If it is more accurate, do you advocate*
*that the community use this instead of the others? I think it is well established here and*
*elsewhere that the inter-kernel differences are small relative to inter-model spread; why are we*
*regularly making new kernels in this case?*
We added more emphasis on the accuracy of this newly generated datasets in the abstract and
conclusion. In short, ERA5 TOA kernels are as good as other datasets but ERA5 surface kernels
show improved performance compared with others (e.g., Figure 10). This is possibly caused by
how the surface kernels are calculated and averaged, e.g., concerning the issues of surface flux
kernels of atmospheric temperature. We also emphasized the importance of the consideration of
surface pressure when vertically integrating the atmospheric contributions.
**Minor Comments**
*• Verbiage: Throughout the paper, I found some of the verbiage to be unnecessarily*
*longwinded. Could "kernel of the surface flux" be the "surface kernel", for example?*
Revised
*• L22: "in" should be "for"*
Corrected.

*• L32: I don't understand what is meant by "inter-kernel bias-induced uncertainty", which appears in slightly modified phrasing in other places as well (L557). Is this just "inter-kernel differences"?*
Corrected.

*• L38: delete "on the other hand"*
 Corrected.

*• L60: suggest also citing the recent work of Chao and Dessler (2021)*
 Added

*• L75: Suggest citing some additional work, some of which includes surface and atmosphere cloud radiative kernels (Zhang et al., 2021; Zhou et al., 2022, 2013)*
Added

*• L81: suggest specifying "largely insensitive"*
Clarified.

*• L83: "are" should be "is"*
Corrected

*• L107: suggest simplifying to "we intercompare"*
Revised

*• L109-111: suggest rephrasing this sentence, which I found hard to parse. Also, you probably want to specify that you are comparing across-model vs across-kernel differences in this sentence (I think)*
Revised

*• L150-152: I'm confused by how you describe the analysis. I thought kernels were constructed using one experiment, performing many calls to the radiative transfer code, each time with a single field / level / location perturbed. This is not how the procedure is described here.*
Clarified.

*• L168-169: Probably want to remind the reader why the factors of 4 and 8 are present in these expressions. It is because the radiation calculations are done 4- or 8-times daily, I think.*
Added.

*• L168: "kernels" should be singular*
Corrected.

*• L190: suggest "upwelling" instead of "outgoing". Also, suggest simplifying to "the kernel is negative"*
Revised.

• *L206: should be "(f,l)" rather than "(g,l)"*
Corrected.
• *L253: "reduce" should be plural*
Corrected.
• *L257: I think you should specify that you are talking about the clear-sky TOA kernel here.*
Added.
• *L336: "by the inconsistency in" should be "by inconsistencies in"*
Corrected.
• *L343-344: could this be simplified to "state-dependency in the kernels"?*
Revised.
• *L354-355: "the" before "interannual" and "cloudiness" is not needed*
Corrected.
• *L359: what is meant be "seasonal SST anomalies" Previously, it is stated that you are*
*examining annual means.*
Revised.
• *L363: "since" should be "in the"*
Corrected.
• *L364: "exemplify" should be "illustrate" or "highlight"*
Corrected.
• *L365: All sky what? Kernels?*
Revised.
• *L370: I think some explanation of this result is warranted. Why does Figure 9e have that*
*structure, wherein some regions that are moister and cloudier have a larger SW WV kernel*
*but some do not (NE Pacific). Also, the panel titles in Figure 9 are a little ambiguous; suggest*
*explicitly stating what is shown in each.*
Corrected.
• *Figure 10: suggest deleting the longitude labels which clutter the figure and seem*
*unnecessary given the provided coastlines.*
We think this is fine.
• *L385-394: More explanation of why you get these results is needed. Also, this is too long of a*
*sentence.*
Revised.

• *L390-394: Is one of the take-aways here that it may be necessary to average over multiple*
*years when constructing kernels? Or at least that one has to be careful not to choose a year*
*with an extreme Nino index or huge sea ice anomalies when constructing kernels? You might*
*consider making this point explicitly.*
Yes, added.
• *L403: missing space between "Table" and "2"*
Corrected.
• *Table 2: "model top level" is not an accurate description of what is reported in that column*
Revised.
• *L412-419: I think more description and motivation for using these experiments is needed.*
*The abrupt-4xCO2 experiment is a fully-coupled experiment whereas piClim-4xCO2 is an*
*atmosphere-only experiment. You should also cite the relevant piClim-4xCO2 experiment*
*description paper (Pincus et al., 2016). I've never seen these two experiments differenced in*
*order to derive the temperature-mediated responses without the confounding effects of rapid*
*adjustments; this is clever although it limits the number of models available to analyze.*
*(Although more than just 6 models are available as far as I can tell.) I suggest explaining*
*these choices a little better. I would also suggest mentioning this methodological difference*
*when coming your values to those of Zelinka et al (2020) – that study used piControl*
*simulations as the baseline and computed abrupt-4xCO2 anomalies and feedbacks differently.*
*It is reassuring that the results of the two approaches agree as well as they do.*
Added.
• *L445-446: The end of this sentence is redundant with previous statements; suggest deleting.*
We think it is fine.
• *L460, L467: small relative to what?*
Added. Compared with the total feedback.
• *L477: Suggest stating the name of the row rather than making the reader count.*
Revised.
• *L478-480: suggest citing some examples to explain how you arrive at these percentage*
*numbers. Are you comparing inter-kernel standard deviations to inter-model standard*
*deviations?*
Revised.
• *L488: these numbers seem misleading, because most feedbacks have roughly the same*
*absolute value of inter-kernel spread; they just vary in the central value. If all feedbacks had*
*the same inter-kernel spread, but one feedback happened to be zero (e.g., if the SW cloud*
*amount feedback perfectly compensated a SW cloud albedo feedback), the inter-kernel spread*
*relative to this would be infinite, but that is not really meaningful.*
Revised.
• *L541: Delete "First of all"*
Deleted
• *L583: This sentence seems to run on and should probably be broken up for clarity.*

Revised.
• *L585-591: this sentence is also way too long and should be broken up*
Revised.
• *L594: "it is especially noticed that" can be deleted*
Deleted.
• *L599-600: suggest making this more concise by removing redundancy*
Revised.
• *L601-602: How could inter-model spread come from inter-kernel spread? Please rephrase.*
Corrected.
• *L603: rephrase to "finding is consistent with previous"*
Revised.
• *L762: specify whether this is an absolute or relative change. I'm pretty sure it is the former.*
Added.
• *L767: not sure what is meant by the last phrase*
Revised.
• *L818: "trickiness" is probably too informal; suggest "challenge"*
Revised.
• *L835: I don't understand what is meant from "and accounting" onward*
Revised.
• *Figure A2: are these SFC or TOA kernels? I assume SFC.*
Yes, added.
• *L854: specify "in these cases"*
Revised.

**References**
*Chao, L.-W., Dessler, A.E., 2021. An Assessment of Climate Feedbacks in Observations and*
*Climate Models Using Different Energy Balance Frameworks. J. Clim. 34, 9763–9773.*
*https://doi.org/10.1175/JCLI-D-21-0226.1*
*Pincus, R., Forster, P.M., Stevens, B., 2016. The Radiative Forcing Model Intercomparison*
*Project (RFMIP): experimental protocol for CMIP6. Geosci. Model Dev. 9, 3447–3460.*
*https://doi.org/10.5194/gmd-9-3447-2016*
*Zelinka, M.D., Myers, T.A., McCoy, D.T., Po-Chedley, S., Caldwell, P.M., Ceppi, P., Klein,*
*S.A., Taylor, K.E., 2020. Causes of Higher Climate Sensitivity in CMIP6 Models.*
*Geophys. Res. Lett. 47, e2019GL085782. https://doi.org/10.1029/2019GL085782*
*Zhang, Y., Jin, Z., Sikand, M., 2021. The Top-of-Atmosphere, Surface and Atmospheric Cloud*
*Radiative Kernels Based on ISCCP-H Datasets: Method and Evaluation. J. Geophys.*
*Res. Atmospheres 126, e2021JD035053. https://doi.org/10.1029/2021JD035053*
*Zhou, C., Liu, Y., Wang, Q., 2022. Calculating the Climatology and Anomalies of Surface*
*Cloud Radiative Effect Using Cloud Property Histograms and Cloud Radiative Kernels.*
*Adv. Atmospheric Sci. 39, 2124–2136. https://doi.org/10.1007/s00376-021-1166-z*
*Zhou, C., Zelinka, M.D., Dessler, A.E., Yang, P., 2013. An Analysis of the Short-Term Cloud*
*Feedback Using MODIS Data. J. Clim. 26, 4803–4815. https://doi.org/10.1175/JCLI-D-*
*12- 00547.1*

---

## Author Comment (AC2)

**Response to Reviewer Comments**

We thank the reviewer for thoughtful and helpful comments. Below are our responses (in regular font) to the comments (in ***bolded italic*** font).

*Reviewer #2:*
*The paper presents a set of newly calculated radiation flux kernels using the ERA5 reanalysis*
*dataset. The authors discuss how the new radiation flux kernels differ from previous ones and*
*how they can be used to improve our understanding of Earth's climate system. Overall, this*
*paper presents a valuable contribution to the field of climate science by providing a new set of*
*radiation flux kernels that can help improve our understanding of Earth's climate sensitivity. I*
*have several major concerns and recommend a major revision.*
*1. In recent years, one of the improvements of radiative kernels is the development of radiative*
*kernels at the surface (SFC) and in the atmospheric column. The kernels at SFC have been*
*calculated not only from reanalysis data but also from observational data (Karmer et al. 2019).*
*Although the ERA5-derived kernels show high consistency with model-based kernels,*
*feedback parameters obtained from model- and reanalysis-based kernels have large*
*discrepancies with observation-based feedback parameters, especially for the cloud feedback*
*(Karmer et al. 2019; Zhang et al. 2021). Would you like to conduct more analysis and add*
*more discussion on the differences in cloud feedbacks derived from various data sources?*
Agreed: we added in the kernel comparison the kernels based on CloudSat dataset (Kramer et al.,
2019) (Figure 3-4) and also in the radiative feedback quantification (Figure 7-10)
*2. Cloud feedbacks are diagnosed using the adjusted cloud radiative effect method by*
*assuming that all-sky decomposition has the same non-closure residual. There are some flaws*
*in the assumption. First, the residual ($res^o$) is introduced during the single variable*
*perturbation or linear decomposition without involving cloud related process. Second, the all-*
*sky decomposition is assumed that has the same non-closure residual with clear-sky ($res^o$*
*$=res^c$). It should be proved before being applied. Once the cloud related processes are*
*introduced, it would be nearly impossible for the non-closure residual in all-sky to be same as*
*the residual in clear-sky. Please reconsider Eqs. 5-6.*
This may be justified as the non-cloud nonlinear effects are comparable in the clear- and all-skies
and the cloud-related terms normally dominate the nonlinear effects in the all-sky
decomposition. We recognize there are inaccuracies in the adjusted cloud radiative forcing
method, although it is the most widely used. This issue is beyond the scope of this paper but
warrants future investigation.
*3. The non-closure residual terms due to nonlinear effect are discussed in Figs. 11 and 12. As*
*shown in Fig. 11, the residual term at the TOA mainly arises from shortwave radiation over*
*regions with abundant sea ice cover. Huang et al. (2021) pointed out that the nonlinear effects*
*are resulted from the coupling effect between the surface albedo and cloud, and between the*
*air temperature and cloud. Given the significant interactive between cloud and other climate*
*variables, it's inappropriate to assume the same residual between all-sky and clear-sky*
*conditions. For the residual term at the SFC (Fig. 12), the magnitude of longwave radiation is*
*comparable to the magnitude of shortwave. There is a lack of necessary discussion of the*
*increase in LW residual at SFC relative to that at TOA.*
See the response above, about the same issue of adjusted CRF method.

We see no strong evidence that surface residual is larger than TOA from figure 7 to 10, although
there may be reasons for this to happen, e.g., because temperature and water vapor feedbacks and
their biases tend to compensate for the TOA but not so for the surface. This is only a speculation
though and would require further investigation to verify.
*4. The most important issue is that what's the contribution of ERA5-based kernel to the*
*radiative kernel method. It's highly consistent with model simulation-based kernel, while*
*model simulation can be applied to more accurate analysis such as diagnostic analysis on the*
*role of dynamic processes in climate response.*
We added notes and discussions on the accuracy of ERA5 kernel in the abstract and conclusion.
In short, the ERA5 TOA kernels are as good as other kernel datasets while for surface kernels,
ERA5 kernels show better performance, in terms of both the radiative sensitivity and radiation
closure test. Model based radiative kernels show good performance in TOA radiation budget
while for surface, they may have some issues, e.g., larger inaccuracies and misattributed surface
contribution (e.g., Figure 10, Figure A2). For observation-based kernel, for example, CloudSat
kernel (Kramer et al., 2019), it also performs well for TOA but not that well for surface. Besides,
satellite observations are subject to the detection of near surface layers and this may lead to some
underestimated radiative sensitivity from the bottom layer for surface kernels. The newly
generated ERA5 show good radiative closure for both TOA and surface and may best facilitate
the analysis of surface energy budget change.
*5. The order of the figures needs to be adjusted. It would be better to cite figures near the*
*context instead of figures far away from the context.*
We reorganize the figures.
*6. In Fig. 6b, the fractional discrepancies of the sensitivity of the TOA SW flux to water vapor*
*in the tropics show six large value centers from the east coast of Africa to the equatorial*
*eastern Pacific. It's hard to understand these large value centers physically. Could you explain*
*it?*
This periodic pattern is caused by CAM3 kernel, likely due to a coarse temporal resolution that
does not well resolve the diurnal cycle of solar insolation (Line 303-304)

*References*
*Kramer, R. J., A. V. Matus, B. J. Soden, and T. S. L'Ecuyer, 2019: Observation-based*
*radiative kernels from CloudSat/CALIPSO. J. Geophys. Res. Atmos., 124, 5431–5444,*
*https://doi.org/10.1029/2018JD029021.*
*Zhang, Y., Z. Jin, and M. Sikand, 2021: The top-of-atmosphere, surface and atmospheric*
*cloud radiative kernels based on ISCCP-H Datasets: Method and evaluation. J. Geophys.*
*Res. Atmos., 126, 1–34, https://doi.org/10.1029/2021JD 035053.*

---

## Author Comment (AC3)

**Response to Reviewer Comments**

We thank the reviewer for thoughtful and helpful comments. Below are our responses (in regular font) to the comments (in **_bolded italic_** font).

*Reviewer #3:*
*In this manuscript, the authors introduce a new set of all-sky and clear-sky, top-of-atmosphere*
*(TOA) and surface radiative kernels, generated with the RRTMG radiative transfer model for*
*5 years of input fields from ERA5 reanalysis data. The authors incorporate these kernels into*
*a more general inter-comparison of the magnitude and structure of existing sets of radiative*
*kernels, highlighting the sensitivity of radiative kernels to the input climate state fiends used to*
*compute them. Along these lines, they also highlight the sensitivity of radiative kernels to*
*interannual variability in the climate state, taking advantage of the fact that the ERA5 kernels*
*have been computed for 5 years of data, over a period with notable changes in ENSO and sea*
*ice coverage.*
*The manuscript is comprehensive and well written. More multi-kernel comparison analysis is*
*certainly needed, so this will be a welcomed addition to the literature. However, I feel the*
*paper suffers in a few ways by trying to balance all three tasks (introduce a new kernel, multi-*
*kernel comparison, sensitivity of kernels to variability) in one paper. For instance, I think the*
*analysis of inter-annual variability is the most valuable part of the work, but the analysis is not*
*well connected to the multi-kernel comparison, and the analysis is not as in depth as it could*
*be. Additionally, it's not clear why we need the ERA5 kernels when they don't seem all that*
*different from the others, and the kernels based on ERA-Interim reanalysis data (ERAi)*
*previously developed by the second author of this paper were also calculated for 5 years of*
*data and could have been used for the inter-annual variability analysis instead. Given the*
*journal, I think justifying the new data product is important. My comments below reflect these*
*concerns and hopefully add some additional, useful explanation to my points. Given these, I*
*think the paper deserves consideration for publication, pending major revisions.*
                                                                                          *Ryan Kramer*
*General*
*1) As noted above, some additional justification for producing the ERA5 kernels is necessary,*
*particularly given that the ERAi kernels exist, which use similar input data and a similar*
*radiative transfer code, also for 5 years of data. For example, is the improvement of RRTMG*
*(ERA5 kernels) over RRTM (the ERAi kernels) large enough to warrant new kernels? Even if*
*so, the sensitivity of kernels to RT is not really a focus of the analysis here. Does ERA5 have*
*more realistic climate state fields than ERAi? The two kernels were computed over different*
*periods, so maybe that is reason to make new kernels? But the period for the ERAi kernels*
*(2008-2012) also has notable swings in ENSO, so I don't quite see the advantage of the later*
*period used for ERA5 kernels.*
*1a) Related, what is the justification for developing radiative kernels from reanalysis*
*when the fields are available from models and observations? Arguably reanalysis offers a*
*happy middle between the two. They may not be pure observations, but they do have the full*
*diurnal cycle that most satellite observations do not have. This may be important for*
*diagnosing feedbacks in models, where the model fluxes are also a response to the full diurnal*
*cycle. But there is also the argument that, in order to diagnose the true feedback, model*
*feedbacks should be diagnosed with a kernel developed from models and observed feedbacks*
*should be diagnosed with appropriate observations. What is the value of reanalysis-based*
*kernels in that context? Some discussion along these lines would be really valuable to a*
*community often confused about what kernels they should be using.*

Following these suggestions, revisions are made to emphasize the strength of this newly
generated ERA5 kernel in the abstract and conclusion. As suggested by the comparison and
figures in the manuscript, ERA5 TOA kernels are as good as other datasets. Though for surface
kernels, it shows better performance than other datasets. These points are added in the revised
manuscript.
Many studies have shown the superior performance of ERA5 compared with other reanalysis,
including its older version - ERAi reanalysis. We used this reanalysis for better representation of
the real atmosphere and recommend using ERA5 kernels for feedback analysis. Compared with
the satellite observation, as the reviewer mentioned, the strength of reanalysis dataset is that it
includes the full diurnal cycle and is not limited by the detection of near-surface layers. For
example, although the CloudSat kernels show great performance for TOA radiation budget, some
of its surface kernels show underestimated strength from the bottom atmospheric layers possibly
due to the difficulty of satellite in detecting low atmosphere information.
It is verified here that the TOA radiative kernels show little discrepancies among the datasets and
use of a reanalysis based kernel can well quantify the radiative feedbacks in GCMs. But the same
cannot be said about the current surface kernels as illustrated by Figure 9 and 10. More
discussion is added in the manuscript to underline this point.
*2) This may be the only example where an "older" (ERAi) and "updated" (ERA5) radiative*
*kernel were developed by the same research group using similar RT codes. This could be a*
*really powerful tool for the multi-kernel comparison analysis and should be exploited here,*
*but has not been yet. If the second author still has access to the ERAi kernel input data, I*
*would like this team to include a more in-depth comparison of the ERAi and ERA5 kernels in*
*the context of the multi-kernel intercomparison. Throughout the current manuscript, the*
*authors highlight examples of large multi-kernel differences, tying them to potential*
*differences in the underlying climate input data. For a given example, is the spread also*
*evident in differences between the ERAi and ERA5 kernels? If so, the authors should analyze*
*the climate input fields directly to reveal specifics about why the kernels differ. A few specific*
*comments in the section below try to prompt this type of analysis. Among other groups, I think*
*this could be extremely useful for the ECMWF developers of ERAi and ERA5, who are always*
*trying to understand the biases and limitations of their product, giving your work exposure to*
*an additional, large community.*
Yes, we added the comparison between ERA5 and ERAi kernel in Figure 5. In general, the
differences between these two kernels are smaller than the interannual variation of ERA5
kernels, except for WV SW kernel, which is largely affected by the difference in cloud and water
vapor fields between ERA5 and EARi.
*3) Section 3.1: The authors should rethink the presentation and the focus of discussion in this*
*section. First, general descriptions of the sign, basic explanation of the causes of the sign, and*
*the zonal-mean vertical structure of kernels are discussed here for the new ERA5 kernels, but*
*these topics have been covered extensively for other kernels and the new kernels don't seem to*
*deviate from that. Therefore, it seems redundant to repeat that information here. This is true*

*even for surface radiative kernels, where the structure and sign were covered by Kramer et al.*
*2019 a and b (see refs below). The description of the horizontal spatial structure of the kernels*
*is newer however, and worthy of focus in this section. Second, the number of figures and*
*figure panels in this section is also overwhelming for the reader and should be consolidated.*
*Given these points, I would instead:*
*-For Figures 1-8, make the first figure or two just the spatial maps of each kernel, with a title*
*for each subplot that describes which kernel we are looking at (e.g. all sky, surface temp,*
*clear-sky SW WV, etc.).*
*-Given their prevalence elsewhere (e.g. Soden et al. 2008; Block and Mauritsen 2013; Kramer*
*et al. 2019a,b, Smith et al. 2021), the latitude-pressure subplots of the ERA5 kernels can be*
*combined and put in supplemental material. -Any subplot currently referring to the*
*intercomparison across existing kernels should be saved for new, separate figures placed in*
*Section 3.2, where that material is discussed in the text.*
*This above list is just a recommendation. I'm sure there are other ways of reorganizing the*
*plots to improve manuscript readability, but some reorganization is necessary.*
*Kramer, R. J., A. V. Matus, B. J. Soden, and T. S. L'Ecuyer, 2019: ObservaDon-Based*
*Radiative Kernels From CloudSat/CALIPSO. JGR Atmospheres, 124, 5431–5444,*
*https://doi.org/10.1029/2018jd029021.*
*Kramer, R. J., B. J. Soden, and A. G. Pendergrass, 2019: Evaluating Climate Model*
*Simulations of the Radiative Forcing and Radiative Response at Earth's Surface.*
*Journal of Climate, 32, 4089– 4102, hTps://doi.org/10.1175/jcli-d-18-0137.1.*
For the completeness of the description, we kept the descriptions of the sign, basic explanation
and the vertical structure of radiative kernels, so that the readers who are not familiar with
radiative kernels have the basic information. We kept the results in all-sky in the main text and
moved the clear-sky result to the supplement for better readability.
*4) It is evident that clouds play an important role in determining the spatial pattern of the all-*
*sky radiative kernels. More description of the type of clouds impacting the kernels would be*
*very useful and novel. For example, cloud vertical extent? Cloud base height (for sfc kernels)?*
*Optical properties? A deeper analysis of the ERA5 cloud fields would be helpful here. And if*
*the fields are still available for the ERAi kernels, even better.*
Cloud information is documented in Line 122 and 128. As cloud fraction and cloud liquid/ice
water content data are from ERA5, they are of the same resolution as other variables (2.5*2.5, 37
level), extending from 1hPa to 1000hPa. Cloud droplet radii are from CERES 3-hourly dataset
(1*1) and then interpolated to the same resolution as ERA5 data.
*5) I think the analysis of inter-annual variability in the kernel is the most interesting*
*contribution of this paper to the literature. It deserves a more prominent place in the title,*
*abstract, and conclusion section.*

Following this suggestion, we strengthened various aspects of the paper concerning the
interannual variability. These responses are detailed below in the responses to the specific
comments related to this topic.
*6) The inter-annual variability analysis feels disjointed from the introduction of the new*
*kernels in Section 3.1, the multi-kernel comparison, and the feedback estimate section. It is*
*evident that the ERA5 kernels are sensitive to inter-annual variability, but does this really*
*matter for overall kernel spread? For instance, the ERA5 all-sky TOA Ts kernel is clearly*
*sensitive to interannual variability in the Eq. Pacific at ~Longtiude 180, but this doesn't*
*appear to be a particularly noteworthy area of inter-kernel differences in fig 1e. It may very*
*well be important, but the analysis at present doesn't offer enough of a connection to make*
*that point. Comparing the inter-annual variability in the ERA5 vs ERAi kernels in more detail*
*may be a useful starting point to help make the connection between this section and the others.*
Both inter-kernel difference and interannual difference of kernel values reflect the dependence of
radiative sensitivity on background atmospheric states. We showed this point using the
comparison among different kernel datasets first and then used the interannual variability of
ERA5 kernel to further show how the change in atmospheric state (e.g., during ENSO or due to
sea ice change) impacts the radiative sensitivity, given that these variables from ERA5 are
available and used in our calculation.
The interannual variability of ERA5 kernel indeed proves this point and further comparison
between ERA5 and ERAi kernel is also added in Figure 5 to compare the changes in kernel value
caused by ENSO. The inter-kernel comparison does not show as much variation in the Central
Pacific as in the ENSO case; this is likely because smaller temperature differences between the
atmospheric datasets used for kernel calculation (e.g., Figure 5g)
*7) After the interesting analysis showing the sensitivity of kernels to inter-annual variability*
*spatially, the feedback analysis in Section 4.2 and 4.3 mostly just focuses on global-mean*
*values. While this type of analysis is valuable in a general sense for kernel users, it doesn't*
*quite fit well with this paper, particularly because the ERA5 kernels don't stand out as being*
*more accurate or unique. Instead, I'd like to authors to focus more on multi-kernel*
*differences in the feedback spatial patterns. Given the large focus on the pattern effect and the*
*influence of regional feedbacks on the global-mean in recent years, I think that could be*
*particularly citeable. We know kernels are generally in agreement in the global-mean*
*(especially for the TOA). But what about for kernel spread in estimates of the regional*
*feedbacks?*
Following this suggestion, we used the spatial root-mean-squares (RMS) of residual terms to
document the spatial biases in each kernel dataset. As indicated by the numbers in Figure 7 and
9, the ERA5 kernels show relatively smaller RMS, especially for surface, compared to many
other kernels. This indicates the EAR5 kernels may be more suitable for surface feedback
quantification. We have clarified these points in the paper.

*8) It is tough to pick out valuable information from your tables of TOA and Surface*
*feedbacks. The authors should turn those into summary figures (e.g. dot plots or scatter plots*
*used by e.g. Smith et al. 2018 supplemental, or Zelinka et al. 2020) where possible.*
Tables are moved to the Supplement and we used figure 8 and 10 to show the inter-kernel and
inter-model feedback spread.
*9) There are now many observation-based kernels in the literature from CloudSat/CALIPSO*
*observations, CERES CCCM products, AIRS, and a bunch of others specific to surface albedo*
*kernels that are not included in the multi-kernel analysis here. I think it's an open question*
*whether these observational kernels should be included in a comparison with model-based*
*kernels or not. The authors should provide a brief reason for not incorporating them (or*
*should include them if they feel its appropriate), especially since the Kramer et al. and*
*Thorsen et al. reference papers are cited in the text.*
Yes, we included the CloudSat kernel for comparison in the revised manuscript.
*Specific or Minor Comments*
*Line 311-312 and Appendix: We discussed similar RT issues regarding the surface*
*temperature kernel for surface fluxes in Kramer et al. 2019 (JClim). The authors can cite*
*and/or refer to it for some additional support. We also argued that, as noted in the author's*
*current Appendix and elsewhere, similar RT issues can bias the lowest level of the surface flux*
*Ta kernel in an equal and opposite manner as the Ts kernel, thereby allowing a kernel like*
*CAM5 to be correct in its estimate of the vertically integrated Temperature feedback, but for*
*the wrong reason. This likely explains why CAM5 and ERA5 kernels agree in Figure A2c and*
*A2d. Your text somewhat gets to this point, but I'd call it out directly as a warning to kernel*
*users: Some kernels may give you the correct temp. feedback for the wrong reason.*
*Presumably this is true for the TOA temperature feedback too, but the contribution from the*
*surface and near surface layers to that vertically integrated feedback are small, so maybe it*
*doesn't matter much?*
*Kramer, R. J., B. J. Soden, and A. G. Pendergrass, 2019: Evaluating Climate Model*
*Simulations of the Radiative Forcing and Radiative Response at Earth's Surface.*
*Journal of Climate, 32, 4089– 4102, https://doi.org/10.1175/jcli-d-18-0137.1.*
Following this suggestion, we added a note to caution this issue in Line 864-865.
*Line 316-319 and more generally: Bright and O'Holleran (2019) and Donohoe et al. (2020)*
*performed nice, detailed comparisons of surface albedo kernels. These papers should be cited*
*and their work should be put into context of the author's own results within the present*
*manuscript. This would also be useful for the section on diagnosing radiative feedback spread,*
*since the authors show that the kernels give quite different results in the poles. Riihela et al.*
*(2021) also did a comparison of surface albedo kernels in the context of sea ice states, and*
*should be cited somewhere in the present manuscript.*
*Bright, R. M., and T. L. O'Halloran, 2019: Developing a monthly radiative kernel for surface*
*albedo change from satellite climatologies of Earth's shortwave radiation budget: CACK*
*v1.0. Geosci. Model Dev., 12, 3975–3990, https://doi.org/10.5194/gmd-12-3975-2019.*

*Donohoe, A., E. Blanchard-Wrigglesworth, A. Schweiger, and P. J. Rasch, 2020: The Effect*
*of Atmospheric Transmissivity on Model and Observational Estimates of the Sea Ice*
*Albedo Feedback. Journal of Climate, 33, 5743–5765, hTps://doi.org/10.1175/jcli-d-19-*
*0674.1.*
*Riihelä, A., R. M. Bright, and K. Anbla, 2021: Recent strengthening of snow and ice albedo*
*feedback driven by Antarctic sea-ice loss. Nat. Geosci., 14, 832–836,*
*https://doi.org/10.1038/s41561-021-00841-x.*

There references are now added.
*Line 316-227 and more generally: Aligning with general comment 4 above, the author's*
*assumption that clouds are the cause of the discrepancies discussed here is likely right, but*
*this has been alluded to before in past work. The author's have a unique opportunity to prove*
*it directly by using the kernel's cloud input data directly in the analysis. Using the ERA5 and*
*ERAi cloud fields can the authors confirm their assumption that cloud fields matter? Or*
*provide a more detailed analysis? Among all the potential sources of kernel differences, clouds*
*seem to warrant deeper investigation.*
We included a comparison between ERA5 and ERAi kernel in Figure 5 and there the difference
in cloud mainly leads to the difference in water vapor SW kernel as this difference only appears
in all-sky (Figure 5l) but not in the clear-sky (Figure S7f).
*Figure 2f: The ERAi and ERA5 kernels are noticeably different below ~800mb. Why?*
Mainly due to the difference in water vapor and air temperature below 800hPa.
*Figure 3e and 3k: There are large standard deviations relative to the magnitude of the ERA5*
*kernels at certain locations within the vertical kernel structure, but for anything above*
*~950mb, the kernel magnitude is small relative to the lowermost atmospheric levels, and likely*
*does not contribute much to the vertically-integrated quantity. A zoomed in version of these*
*plots, highlighting the important standard deviation across kernels in the lowermost levels,*
*would be informative.*
We would like to keep the current vertical range as the discrepancy in the South Pole region
extents to about 300hPa.
*Figure 3f and 3l. Why is the ERAi kernel so much larger at the lowest levels near the surface*
*than the ERA5 kernel (and the other kernels)? This is true for both all-sky and clear-sky.*
*Could vertical resolution of the kernels be playing a role? We discuss the potential influence*
*of resolution in the Appendix of Kramer et al. (2019, JClim).*
Yes, this is due to the vertical resolution and how the air temperature perturbation is added.
*Interestingly, there is also a fairly large difference between ERA5 and ERAi in the all-sky LW*
*WV kernel for surface fluxes (figure 5L), but it only shows up in the all-sky kernel, not the*
*clear-sky. Does that suggest clouds matter more for the LW WV kernel in explaining*

*differences than they do for the LW Ta kernel, relative to other potential sources of kernel*
*spread?*
This is an interesting observation and hypothesis. The comparison however may be obscured by
the averaging issues noted in the Appendix. We tried to not speculate here.
*Line 368: Is this the sensitivity of the surface to the vertically integrated kernel change? Only*
*the sensitivity to a certain level of WV change? It is not obvious from the figure 9 caption*
*either. Some rethinking of how the kernels are described in the text here and elsewhere would*
*be helpful. Maybe shorthand acronyms or some other naming convention would be helpful.*
It is relative to the vertically integrated kernel. Caption is revised accordingly.
*Plot 9: I really like this figure. I'm not sure anyone else has shown the sensitivity of these*
*temperature and water vapor kernels to variability spatially yet. But I think it would be helpful*
*to go one step further and show what level of WV and cloud variability is impacting the*
*temporal variability of the kernels most. And does that particular level help explain the multi-*
*kernel spread in those kernels, evident in Figure 1-8? Or is inter-annual variability not*
*enough to explain the kernel spread? This comments connects with my general comment #6*
*above.*
Following this suggestion, we added Figure S8 to show the vertical distribution of water vapor,
cloud profiles and also water vapor kernels. In the ENSO case, the relatively weaker water vapor
LW kernel in the Central Pacific (Figure 5e) is contributed from almost the whole troposphere
(Figure S8c), possibly caused by the increase of cloud cover in the upper troposphere (Figure
S8b). For the difference between ERA5 and ERAi kernel, the vertically integrated difference in
water vapor SW kernel (Figure 5l) is mainly contributed from the mid-to-low troposphere
(Figure S8f), which also corresponds to the discrepancies noticed in Figure 4i, and is partially
due to the increase of cloud cover in mid-troposphere (Figure S8e).
In summary, the interannual variability contribute to but does not fully explain the inter-kernel
differences as shown in Figure 3 and 4, but both of them demonstrate the state-dependency of
radiative kernels. We clarified these points in the paper.
*Plot 9: The TOA SW WV kernel is a potentially interesting case where the largest multi-kernel*
*spread (tropics around 500mb in Figure 6k) sits above the level at which the ERA5 kernel is*
*the strongest (closer to 800mb in Figure 6j). Building on my comment above, how does the*
*importance of inter-annual variability play into the large kernel spread at this ~500mb level?*
*And does the spread at 500mb actually matter much for the vertically-integrated quantity?*
*This level of detail could be useful for e.g. modeling centers trying to connect TOA biases to*
*particular biases in their climate states.*
As shown by Figure S8, it suggests that the interannual variation partly explains the
discrepancies among kernel datasets. The difference of cloud field between ERA5 and ERAi
suggest that the discrepancies in WV SW TOA kernel in all-sky are mainly caused by the cloud.

*Figure 10: I struggle to see spatially where the variability in water vapor is having an effect on the kernels shown in the other subplots. Maybe only for the NE coast of Greenland? Should cloud changes be shown in this plot instead? I'd give more detailed evidence about why water vapor is important here.*

Replaced it with cloud cover.

*Line 412-419: Some explanation of why you need both abrupt4xCO2 and piCLim-4xCO2 (e.g. to remove rapid adjustments) is needed, since most people just use abrupt4xCO2 with Gregory regression to get feedbacks.*

Yes, this is to remove the rapid adjustment. Explanation was added in Line 437.

 *Line 436 and Equation 5: What does the clear-sky residual term mean physically and why does it matter for computing cloud feedback? Although the author's math in Equation 6 works out to be the same as what everyone else uses, I think the way they've introduced this method, and terminology used, is less common. Some extra detail would be helpful.*

The clear-sky residual term means the unexplained part by kernel method. In adjusted cloud radiative forcing method, such a non-closure term is actually attributed to the cloud feedback.

*Line 463-464: Block and Mauritsen (2013) can be cited here for their nice discussion and analysis of the non-linearity of the surface albedo kernel in 4xCO2 runs.*
*Block, K., and T. Mauritsen, 2013: Forcing and feedback in the MPI-ESM-LR coupled model under abruptly quadrupled CO2. J. Adv. Model. Earth Syst., 5, 676–691, https://doi.org/10.1002/jame.20041.*

Added.

*Last Column Table 3: The multi-model values differ somewhat across the different kernels, but not the associated standard deviations, which look to be essentially the same for all rows of the column. Does this suggest the kernels can estimate feedbacks differently in a systematic manner across all models, but they do not necessarily estimate the magnitude of the feedback spread differently? In other words, the kernel may get the model-mean feedback value wrong, but the spread in feedbacks correct? This is worth noting if so.*

As feedbacks are calculated by the product of radiative kernels ($K_X$) and the anomalies ($\Delta X$), when calculating the standard deviation of feedbacks among the models by the same radiative kernels, it is the variation of $\Delta X$ among models that matters (as all models use the same $K_X$) and I think that's why different kernel datasets show a close multi-model standard deviation.

*Line 487-491: Since you are not using cloud radiative kernels, the inter-kernel differences in cloud feedback must come from the difference between all-sky and clear-sky kernels (cloud masking). Can you identify which of the kernel terms is the culprit? From a related discussion see text around figs 9-11 in Kramer et al. (2019, JGR), for example.*

*Kramer, R. J., A. V. Matus, B. J. Soden, and T. S. L'Ecuyer, 2019: Observation-Based Radiative Kernels From CloudSat/CALIPSO. JGR Atmospheres, 124, 5431–5444, https://doi.org/10.1029/2018jd029021.*

In our calculation, we found that the inter-kernel differences in cloud LW feedback are almost equally contributed from Ta, Ts and WV, and in cloud SW feedback are contributed more from the albedo. As no dominant contributor or general feature is found, we chose to make no specific additional comment here.

*Line 769-793: This is a really nice description of how you develop the water vapor kernel. I'd highlight in the main text that you have included this section in appendix. I think many kernel users and developers are looking for a description like this and will turn to it in the future. There are often question about this calculation.*

We chose to keep the technical details in the appendix.

---

## Referee Report (RR1)

**Review of Radiative sensitivity quantified by a new set of radiation flux kernels based on the ERA5 reanalysis**
**By Huang and Huang**
**essd-2022-474**

**Summary**
The authors have improved the paper relative to its previous version and I am mostly satisfied with the changes they have made in response to my and other reviewers' comments. There remain a few places where further revisions are needed, which I detail below.

**Specific Comments**
- When making use of these kernels, I initially struggled to get good closure at the surface. The issue is that the surface temperature and humidity kernels peak at the lowest atmospheric level, but in many models this level is below ground. Thus even if the kernel is nonzero and large, the radiative impact is zero because that level has no change in temperature or humidity (because it is underground). The solution I found was to set the atmospheric temperature and humidity values equal to their surface values anywhere that they were zero or undefined at the lowest levels of the atmosphere. This correction ensures that there is something nonzero to multiply the kernel by at the near-surface level where the kernel peaks. I think you may need to provide this methodological detail somewhere in the paper in order for people to correctly implement these kernels.
- Figure 5: I don't feel as though the results shown in the right column of this figure are adequately explained. I think the statement on L370 is incorrect: Rather, the figure indicates that the negative surface temperature kernel has strengthened in ERA5. Why has this happened? I cannot rationalize this from looking at the changes in Figure 5g,h,i. I would have thought the moister atmosphere might weaken the surface temperature kernel (the opposite of what happens). I also am not sure what is being referred to on L373 regarding the linkage between the discrepancy noted in Figure 4i and the SW WV kernel results. Please elaborate on this.
- Figure 6: This is also not explained particularly well. It is stated on L391-392 that "the reduction of sea ice in the Arctic region leads to a significant decrease of radiative sensitivity to surface albedo". If this statement is taken at face value, one would expect panels d and f to look like mirror images of panel a, but there is little correspondence at all. But I don't think there is any reason to expect the albedo kernel to depend on surface albedo since it is defined as the SW impact of a 1% increase in albedo. This kernel mainly varies with insolation and cloud cover. So the change in total cloud cover (panel b) actually explains most of the geographical structure in the change in surface albedo kernel (rather than "also contributing" as the authors state).
- Figures 8: I find it very hard to reconcile the very small SW residuals in Figure 8 with the quite substantial zonal mean residuals shown in the bottom row of Figure 7. Please double check this calculation. I also suggest removing the substantial white space at the top and bottom of this figure, since no values extend above about 2 W/m2/K or below about -3.5 W/m2/K. In the caption, "list" should be "listed", and "pentagrams" should probably be "stars."

- Abstract: There is no mention of the analysis regarding dependence of the kernel on mean state (as examined in Section 3.3)
- L22: seems odd to not mention climate change in addition to variability here.
- L65: should be "approximate"
- L81: you may also consider citing Figure S2 of Zelinka et al (2020)
- L87: I don't think "calls into question" is the right phrase here. Perhaps "…this warrants investigating whether…"
- L108: should be "set" (singular)
- L147-156: This methodological description is still awkward. You are not performing separate simulations, right? There is one control simulation, and within that simulation you perform multiple radiative calculations, each time with a small perturbation in a field. I think "simulations" should be replaced with "calculations" in most cases.
- L152: should be "calculate **the** radiative"
- L172-173: Many analyses in the paper use monthly resolved kernels, but I don't think this is stated when it occurs.
- L180: I think you should explain explicitly what atmospheric kernels are here, since it may not be obvious what they are or how they are computed
- Figure 1 and others: Need to notify the reader that the colorbar ranges vary among panels
- L258: I understand why the authors refer to the Kramer et al (2019) kernels as CloudSat/CALIPSO, but this could possibly confuse readers who might think they are cloud radiative kernels. My understanding is that these make use of thermodynamic fields from ECMWF, so they actually use similar inputs as the ERA kernels developed here. I think a brief clarification of what these kernels are is warranted to avoid confusion.
- Table 1: Suggest renaming the third column as "Vertical levels" or something, since the resolution is not shown
- L289: should be "fractional"
- L299,308,404 and elsewhere: The word "biases" still shows up in the revision even though these are not biases.
- L323: 10% of what? Please specify
- L426-427: commas are not needed after the experiment names, and "and" should be inserted after "1850,"
- L432: Rather than "following the previous studies" this methodology deviates in a fairly significant way from Zelinka et al and Smith et al (and most studies that use abrupt-4xCO2 experiments). Namely, the conventional way is to difference piControl and abrupt over the duration of the abrupt run, compute annual mean values, and regress on annual mean surface temperature anomalies (the Gregory method). The method used here is quite different and needs to be explained and motivated better. Note – I am not criticizing the method. I just want you to explain and motivate it better, and to delete the phrase about it following previous studies. One nice motivation is that it obviates the need to worry about rapid adjustments.
- L441: "notre" should be "note"
- L447: delete "atmospheric" since it includes surface temperature and albedo

- L449: Somewhere in this section you need to note that the kernels and the climate fields they are multiplied with are at monthly resolution
- L460-471: For any casual reader, this description of how to compute cloud feedbacks is probably inadequate and bewildering. As another reviewer noted, the math ends up the same, but the physical connection of the equations to how it relates to clouds is lost. Suggest re-doing this (or appending discussion onto it), perhaps adhering more closely to Eqs. 22-25 in Soden et al (2008).
- Figure 7: The figure panels are too small, partly because there is so much redundant information that is repeated. All colorbars are identical, so there is no need to show them near each panel – this would clear up a lot of space. You could also label each row once and each column once rather than putting a title on each panel.
- Figure 10: The ERA5 kernels produce anomalously large Ts and Ta feedbacks relative to the other kernels, but I don't think this is discussed at all. Please discuss.
- L579: should be "containing." Also I would suggest noting in this section (rather than earlier in the text) that the multi-kernel dataset is also provided at this link. Thank you for providing this.
- L587: I don't think "including the kernel values" is needed here as this is obvious
- L594: should be "Antarctic" or "over Antarctica"
- L598, L617: 30%/10% of what?
- L600: suggest pointing the reader to the Appendix here.
- L605: I think you should say this "might explain" the discrepancies, since you have not established this across kernels (which also differ in other ways including radiative transfer codes)
- L618 and elsewhere: The word "affirm" appears 10 times in the manuscript; suggest using a synonym occasionally.
- L807: "the multiply of" is not the correct phrasing

**References**

Kramer, R. J., Matus, A. V., Soden, B. J., & L'Ecuyer, T. S. (2019). Observation-Based Radiative Kernels From CloudSat/CALIPSO. *Journal of Geophysical Research: Atmospheres*, *124*(10), 5431–5444. https://doi.org/10.1029/2018JD029021

Soden, B. J., Held, I. M., Colman, R., Shell, K. M., Kiehl, J. T., & Shields, C. A. (2008). Quantifying Climate Feedbacks Using Radiative Kernels. *J. Climate*, *21*, 3504–3520. https://doi.org/10.1175/2007JCLI2110.1

Zelinka, M. D., Myers, T. A., McCoy, D. T., Po-Chedley, S., Caldwell, P. M., Ceppi, P., et al. (2020). Causes of Higher Climate Sensitivity in CMIP6 Models. *Geophysical Research Letters*, *47*(1), e2019GL085782. https://doi.org/10.1029/2019GL085782

---

## Author Response (AR2)

**Response to Reviewer Comments**

We thank the Editor and reviewers for their additional comments. Below are our responses (in regular font) to their comments (in ***bolded italic*** font).

***Editor:***
*Dear Authors,*
*Both reviewers report that the manuscript has been improved and has the potential to become a valuable contribution in documenting the dataset.*
*Both reviewers have a number of specific revisions that they suggest implementing.*
*I would like to thank the reviewers for their efforts.*

All addressed.

	*Report #1*
	*Suggestions for revision*
*Review of Radiative sensitivity quantified by a new set of radiation flux kernels based on the*
*ERA5 reanalysis*
*By Huang and Huang*
*essd-2022-474*
*Summary*
*The authors have improved the paper relative to its previous version and I am mostly satisfied*
*with the changes they have made in response to my and other reviewers' comments. There*
*remain a few places where further revisions are needed, which I detail below.*
*Specific Comments*
*When making use of these kernels, I initially struggled to get good closure at the surface. The*
*issue is that the surface temperature and humidity kernels peak at the lowest atmospheric*
*level, but in many models this level is below ground. Thus even if the kernel is nonzero and*
*large, the radiative impact is zero because that level has no change in temperature or humidity*
*(because it is underground). The solution I found was to set the atmospheric temperature and*
*humidity values equal to their surface values anywhere that they were zero or undefined at the*
*lowest levels of the atmosphere. This correction ensures that there is something nonzero to*
*multiply the kernel by at the near-surface level where the kernel peaks. I think you may need*
*to provide this methodological detail somewhere in the paper in order for people to correctly*
*implement these kernels.*
We thank the reviewer for this suggestion and added on Line 969 the following texts:
"To illustrate this issue in an example, consider a location (latitude-longitude grid point)
where the surface pressure is 960 hPa in a GCM and the lowermost level of non-zero value of
ERA5 air temperature kernel is located at 975 hPa. Had the air temperature change been set to
zero or NaN value due to the GCM ground level being above 975 hPa, the contribution to the
surface radiation change from the air temperature change in the bottom layer of the atmosphere
would not be included, which may lead to a biased quantification of the feedback. We
recommend interpolating the air temperature changes from the GCM vertical profile to the kernel
vertical profile, using surface values to replace the missing levels (e.g., the 975 hPa level in the
above example) before multiplying with the kernel values, when computing the feedbacks of air
temperature and water vapor."
*Figure 5: I don't feel as though the results shown in the right column of this figure are*
*adequately explained. I think the statement on L370 is incorrect: Rather, the figure indicates*
*that the negative surface temperature kernel has strengthened in ERA5. Why has this*
*happened? I cannot rationalize this from looking at the changes in Figure 5g,h,i. I would*
*have thought the moister atmosphere might weaken the surface temperature kernel (the*
*opposite of what happens). I also am not sure what is being referred to on L373 regarding the*

*linkage between the discrepancy noted in Figure 4i and the SW WV kernel results. Please elaborate on this.*

To clarify these results, we added the following text at Line 383:

"Although the total column water vapor and total cloud cover are higher in the ERA5 (Figure 5h and i), their differences are complex and vertically non-uniform (Figure S8 d and e), which leads to a slight strengthening of surface temperature kernels compared with ERAi (Figure 5j)."

*Figure 6: This is also not explained particularly well. It is stated on L391-392 that "the reduction of sea ice in the Arctic region leads to a significant decrease of radiative sensitivity to surface albedo". If this statement is taken at face value, one would expect panels d and f to look like mirror images of panel a, but there is little correspondence at all. But I don't think there is any reason to expect the albedo kernel to depend on surface albedo since it is defined as the SW impact of a 1% increase in albedo. This kernel mainly varies with insolation and cloud cover. So the change in total cloud cover (panel b) actually explains most of the geographical structure in the change in surface albedo kernel (rather than "also contributing" as the authors state).*

We agree that clouds also explain the surface albedo kernel value differences. However, we would like to point out that surface albedo kernel value does depend on the surface albedo value, as discussed by Huang et al. (2021, c.f. the univariate nonlinearity discussion there, e.g., Fig 3 and 6), due to the multiple-scattering between the surface and the atmosphere.

Huang, Y., Huang, H., & Shakirova, A. (2021). The Nonlinear Radiative Feedback Effects in the Arctic Warming. *Frontiers in Earth Science*, *9*, 693779.

To clarify these results, we revised the relevant texts at Line 408:
"In the sea ice loss case, the reduction of sea ice in the Arctic region (Figure 6a) leads to a significant decrease of radiative sensitivity to surface albedo in the areas with noticeable sea ice retreats (Figure 6d and f), with the maximum difference exceeding 30% of the radiative kernel value, because of the nonlinear dependency of the reflected solar radiation on the surface albedo (e.g., see Huang et al., 2021b, Fig. 3 and Fig. 6). The cloud cover changes also contribute to changes in surface albedo kernel values due to the coupling effect between cloud and surface albedo (see Huang et al., 2021b), which for example is seen in the Siberia and to the west coastline of Europe."

*Figures 8: I find it very hard to reconcile the very small SW residuals in Figure 8 with the quite substantial zonal mean residuals shown in the bottom row of Figure 7. Please double check this calculation. I also suggest removing the substantial white space at the top and bottom of this figure, since no values extend above about 2 W/m2/K or below about -3.5 W/m2/K. In the caption, "list" should be "listed", and "pentagrams" should probably be "stars."*

We verified the results are correct. This is because the area-weighting limits the effect of the high latitude biases on the global mean.

We revised the figure according to the reviewer's suggestions.

***Abstract: There is no mention of the analysis regarding dependence of the kernel on mean***
***state (as examined in Section 3.3)***

Added.

***L22: seems odd to not mention climate change in addition to variability here.***
***L65: should be "approximate"***
***L81: you may also consider citing Figure S2 of Zelinka et al (2020)***
***L87: I don't think "calls into question" is the right phrase here. Perhaps "...this warrants***
***investigating whether..."***
***L108: should be "set" (singular)***

All corrected.

***L147-156: This methodological description is still awkward. You are not performing separate***
***simulations, right? There is one control simulation, and within that simulation you perform***
***multiple radiative calculations, each time with a small perturbation in a field. I think***
***"simulations" should be replaced with "calculations" in most cases.***

Revised.

***L152: should be "calculate the radiative"***

Corrected.

***L172-173: Many analyses in the paper use monthly resolved kernels, but I don't think this is***
***stated when it occurs.***

Added.

***L180: I think you should explain explicitly what atmospheric kernels are here, since it may***
***not be obvious what they are or how they are computed***

Clarified.

***Figure 1 and others: Need to notify the reader that the colorbar ranges vary among panels***

Added.

***L258: I understand why the authors refer to the Kramer et al (2019) kernels as***
***CloudSat/CALIPSO, but this could possibly confuse readers who might think they are cloud***
***radiative kernels. My understanding is that these make use of thermodynamic fields from***

*ECMWF, so they actually use similar inputs as the ERA kernels developed here. I think a brief clarification of what these kernels are is warranted to avoid confusion.*

Clarified.

*Table 1: Suggest renaming the third column as "Vertical levels" or something, since the resolution is not shown*
*L289: should be "fractional"*

Both corrected.

*L299,308,404 and elsewhere: The word "biases" still shows up in the revision even though these are not biases.*

All corrected and we double-checked other "biases"/"biased" in the manuscript and they all now represent the proper meaning.

*L323: 10% of what? Please specify*

Specified.

*L426-427: commas are not needed after the experiment names, and "and" should be inserted after "1850,"*

Corrected.

*L432: Rather than "following the previous studies" this methodology deviates in a fairly significant way from Zelinka et al and Smith et al (and most studies that use abrupt-4xCO2 experiments). Namely, the conventional way is to difference piControl and abrupt over the duration of the abrupt run, compute annual mean values, and regress on annual mean surface temperature anomalies (the Gregory method). The method used here is quite different and needs to be explained and motivated better. Note – I am not criticizing the method. I just want you to explain and motivate it better, and to delete the phrase about it following previous studies. One nice motivation is that it obviates the need to worry about rapid adjustments.*

The following texts are added at Line 453 to clarify this, following the reviewer's suggestion:

"To exclude the effect of rapid adjustments, the radiative feedbacks in this study are measured using the difference of feedback variables between the abrupt4xCO2 and piClim-4xCO2 experiments and vertically integrated from the surface to model top. Note that these treatments are different from some other studies, e.g., Zelinka et al., 2020, which used piControl simulation as the climatology baseline and vertical integration from the surface to the tropopause, although the quantitative differences in the diagnosed global mean feedback values are small."

*L441: "notre" should be "note"*
*L447: delete "atmospheric" since it includes surface temperature and albedo*

Both corrected.
*L449: Somewhere in this section you need to note that the kernels and the climate fields they*
*are multiplied with are at monthly resolution*
Added.
*L460-471: For any casual reader, this description of how to compute cloud feedbacks is*
*probably inadequate and bewildering. As another reviewer noted, the math ends up the same,*
*but the physical connection of the equations to how it relates to clouds is lost. Suggest re-doing*
*this (or appending discussion onto it), perhaps adhering more closely to Eqs. 22-25 in Soden et*
*al (2008).*
The following explanations are added on Line 491 to clarify our formulation:
"It is worth noting that $\Delta R_c$ measured according to Eq. (6) is essentially the part of total radiation
change not explained by the non-cloud feedbacks and is equivalent to the other formulations of
the adjusted cloud radiative effect method (e.g., Shell et al. 2008; Soden et al., 2008). Interested
readers can refer to, for example, Huang (2013) for a detailed formulation and explanations of
the method."
*Figure 7: The figure panels are too small, partly because there is so much redundant*
*information that is repeated. All colorbars are identical, so there is no need to show them near*
*each panel – this would clear up a lot of space. You could also label each row once and each*
*column once rather than putting a title on each panel.*
We reorganize the panels and now use only one common colorbar in each column, but keep the
title on each panel as the RMS has to be shown there.
*Figure 10: The ERA5 kernels produce anomalously large Ts and Ta feedbacks relative to the*
*other kernels, but I don't think this is discussed at all. Please discuss.*
The following texts are added in Line 584 and Line 900 to discuss this result:
"The sum of air temperature and surface temperature feedbacks shows better consistency
compared with the respective components (except for the HadGEM3 kernel), and the respective
air temperature and surface temperature feedbacks quantified by the ERA5 kernel are stronger
than the results from the other kernels. These discrepancies are due to the reason discussed in the
Appendix – a possibly wrong quantification of surface temperature effect."
"As a result, in Figure 10, we see stronger air temperature and surface temperature feedbacks
quantified from ERA5 kernels than those from other kernels and in Table S4, we can only report
the sum of surface and air temperature feedbacks."

*L579: should be "containing." Also I would suggest noting in this section (rather than earlier in the text) that the multi-kernel dataset is also provided at this link. Thank you for providing this.*

Added.

*L587: I don't think "including the kernel values" is needed here as this is obvious*
*L594: should be "Antarctic" or "over Antarctica"*

Both corrected.

*L598, L617: 30%/10% of what?*

Added.

*L600: suggest pointing the reader to the Appendix here.*

Added.

*L605: I think you should say this "might explain" the discrepancies, since you have not established this across kernels (which also differ in other ways including radiative transfer codes)*

Corrected.

*L618 and elsewhere: The word "affirm" appears 10 times in the manuscript; suggest using a synonym occasionally.*

Modified.

*L807: "the multiply of" is not the correct phrasing*

Corrected.

*References*
*Kramer, R. J., Matus, A. V., Soden, B. J., & L'Ecuyer, T. S. (2019). Observation-Based Radiative Kernels From CloudSat/CALIPSO. Journal of Geophysical Research: Atmospheres, 124(10), 5431– 5444. https://doi.org/10.1029/2018JD029021*
*Soden, B. J., Held, I. M., Colman, R., Shell, K. M., Kiehl, J. T., & Shields, C. A. (2008). Quantifying Climate Feedbacks Using Radiative Kernels. J. Climate, 21, 3504–3520. https://doi.org/10.1175/2007JCLI2110.1*
*Zelinka, M. D., Myers, T. A., McCoy, D. T., Po-Chedley, S., Caldwell, P. M., Ceppi, P., et al. (2020). Causes of Higher Climate Sensitivity in CMIP6 Models. Geophysical Research Letters, 47(1), e2019GL085782. https://doi.org/10.1029/2019GL085782*

*Reviewer #2:*
*Suggestions for revision*

*The authors have thoroughly addressed my comments, revising and adding valuable analysis to the sections I had concerns about. I recommend the manuscript for publication after the authors address a few minor comments below:*

*Line 104-105: In their reviewer response the authors note that many studies have shown the superiority of ERA5, including over ERAi. It would be helpful to cite those papers after "…which demonstrates superior 105 accuracy in the quantification of various atmospheric states"*

Added (Line 110).

*Line 405-407: What if someone wants to use radiative kernels specifically to diagnose radiative feedbacks associated with ENSO? Does this recommendation still apply? Or should they use a kernel that was derived from an El Nino/La Nina year? I wonder if this recommendation is specifically geared towards users who want to compute global feedbacks for long-term climate change?*

We agree the complexity of the situations should be considered when choosing kernels. Here, we mean to advise on how to capture the "mean" radiative sensitivity. This is clarified with the following texts (Line 424):

"If only one year's atmospheric profiles are used to generate radiative kernels, we recommend selecting a year without significant anomalies in atmospheric states, e.g., due to El Nino or severe sea ice loss, so that the computed kernel values better represent the radiative sensitivity climatology."

*Line 516-519: The authors refer to e.g. air temperature and water vapor feedbacks as the source of inter-kernel spread in cloud LW feedback (and albedo for cloud SW feedback). To avoid confusion, they should be a bit more specific mention they are referring to the "cloud masking" terms in the cloud feedback calculation i.e., the difference between all-sky and clear-sky temperature feedback, water vapor feedback, etc.*

Clarified following the reviewer's suggestion (Line 542).

*Figure 9 caption should say similar to figure 7 not figure 8*

Corrected.